# DIFFUSION MODELS FOR MULTI-MODAL GENERATIVE MODELING

**Changyou Chen**[1,2]  **Han Ding**[2]  **Bunyamin Sisman**[2]  **Yi Xu**[2]
**Ouye Xie**[2]  **Benjamin Yao**[2]  **Son Tran**[2]  **Belinda Zeng**[2]
[1]University at Buffalo  [2]Amazon

## ABSTRACT

Diffusion-based generative modeling has been achieving state-of-the-art results on various generation tasks. Most diffusion models, however, are limited to a single-generation modeling. Can we generalize diffusion models with the ability of multi-modal generative training for more generalizable modeling? In this paper, we propose a principled way to define a diffusion model by constructing a unified multi-modal diffusion model in a common *diffusion space*. We define the forward diffusion process to be driven by an information aggregation from multiple types of task-data, *e.g.*, images for a generation task and labels for a classification task. In the reverse process, we enforce information sharing by parameterizing a shared backbone denoising network with additional modality-specific decoder heads. Such a structure can simultaneously learn to generate different types of multi-modal data with a multi-task loss, which is derived from a new multi-modal variational lower bound that generalizes the standard diffusion model. We propose several multi-modal generation settings to verify our framework, including image transition, masked-image training, joint image-label and joint image-representation generative modeling. Extensive experimental results on ImageNet indicate the effectiveness of our framework for various multi-modal generative modeling, which we believe is an important research direction worthy of more future explorations.

## 1 INTRODUCTION

The field of artificial intelligence (AI) has witnessed significant advancements in generative modeling, leading to remarkable progresses such as DALL-E (Ramesh et al., 2022) and GPT-4 (OpenAI, 2023). The generative AI paradigm enables the learning of transitions from simple to complex distributions, such as from a standard Gaussian distribution to a high-dimensional image distribution. Compared to discriminative learning, generative mechanisms can arguably prioritize the overall structures of the data, offering better data fitting and potential robustness to data noise. However, while real-world applications often involve data of multiple types (multi-modal), including images, video, text, and labels, most existing generative models primarily focus on generating a single data type or modality. Notably, the diffusion model (Sohl-Dickstein et al., 2015; Ho et al., 2020), a state-of-the-art generative model, has been independently developed for generating image, text, audio, and label data (Dhariwal & Nichol, 2021; Li et al., 2022b; Liu et al., 2023; Han et al., 2022). *Can we design a principled way to enable joint modeling and generating multi-modal data within the diffusion-model framework?*

Furthermore, leveraging multi-modal information through learning from multiple tasks and data sources has proven to be highly effective to learn generalized representations. Prominent examples include the ALBEF and BLIP models, which jointly learns from multi-modal data to match image and text (Li et al., 2021; 2022a; 2023), and the BERT model, which benefits from multi-task training such as masked token prediction and consecutive sentence prediction (Devlin et al., 2019b). *Can we adopt a similar setting to leverage multi-modal data and losses into the diffusion-model framework, so as to better integrate shared information among tasks for better generative modeling?*

In this paper, we present our initial endeavor towards this goal by introducing the multi-modal diffusion model with multi-task learning, referred to as *MT-Diffusion*. MT-Diffusion enables simultaneous modeling and generation of multi-modal data with a unified diffusion model. By multi-task, we emphasize that MT-Diffusion is designed to 1) simultaneously generate multi-modal data (potentially

Figure 1: Illustration of the proposed MT-Diffusion on two modalities. The diffusion process is defined in a shared *diffusion space* for all modality data, which are transformed from the modality-specific encoders. The forward nosing process includes a forward aggregation step that integrates information from *multi-modal data*, and the reverse denosing component transforms the diffusion space back to the task-specific data spaces with learnable decoders through a *multi-task loss*.

heterogeneous such as images and labels) within a unified model; and 2) seamlessly integrate multi-task learning losses into the diffusion framework in a principled manner, supported by theoretical foundations. Our multi-task setting is versatile and applicable to numerous practical scenarios. It is worth noting that, as an initial investigation to multi-modal diffusion models, we only focus on two modalities to demonstrate the promise of the research direction, while leaving training with more modalities as interesting future work. Particularly, we construct several practical multi-task generative learning scenarios in experiments to demonstrate the effectiveness of our framework:

- **Image transition**: We consider jointly modeling multi-modal data, such as images and the corresponding semantic segmentation masks, by learning to generate both in the reverse process within our MT-Diffusion framework. We design this task as a synthetic experiment to qualitatively demonstrate the ability of our model on small-scaled datasets.

- **Masked-image training**[*]: Motivated from the previous success on masked-language pre-training such as BERT (Devlin et al., 2019a) in language modeling, we propose to combine a pure generation task with a masked-image generation task for generative training. We demonstrate on the ImageNet dataset that our model can be more efficient in training a generative model, and can converge to a point comparable to (if not better than) the heavily tuned single-task diffusion model in terms of generative image quality. Furthermore, it can simultaneously obtain for free a great image-restoration ability for masked image recovery.

- **Joint image-label generation**: We jointly model images and the corresponding labels by learning to generate both with our MT-Diffusion. We demonstrate on the ImageNet dataset that one can achieve better classification accuracy compared to pure supervised training.

- **Joint image-representation generation**: We also investigate simultaneously learning to generate images and representations (*e.g.*, CLIP representations (Radford et al., 2021)) with MT-Diffusion. As this is a larger-scale setting based on stable diffusion, we only provide qualitative results to demonstrate the ability of our model to generate high-quality images from text, while leaving more detailed investigations as interesting future work.

Our solution for these multi-modal generation problems is a novel generalization of the standard diffusion model, designed to handle data from multiple modalities through both innovative algorithm and architecture designs in the diffusion forward and reverse processes. Our general idea is illustrated in Figure 1. In the forward process, multi-modal/multi-task data are first aggregated through some well-designed mechanisms (details in Section 2.2.2) so that the aggregated information can be conveniently applied to the forward noising operation of a diffusion model. To deal with potentially heterogeneous data, an effective encoder architecture design is proposed to encode *multi-modal data* into a shared *diffusion space*. In the reverse process, we propose to extend the original U-Net architecture[†] in diffusion models to simultaneously reconstruct the multi-modal data from different tasks. To this end, modality-specific decoder heads are designed to be attached to the U-Net architecture to decode the diffusion latent code back to multi-modal data spaces. The forward and reverse processes are then integrated within the diffusion mechanism, leading to a loss derived from a new multi-task evidence lower bound (ELBO), as a *multi-task loss*. Extensive experiments on the aforementioned problems are conducted to verify the effectiveness of our framework, demonstrating that our model can achieve simultaneous generation without hurting individual task performance, a promising generalization of the standard diffusion model for multi-modal generative learning.

---

[*]The recent DiffMAE (Wei et al., 2023) is fundamentally different from ours. It is a standard conditional diffusion model to denoise the pre-masked region; ours models both image and mask generation.

[†]The default U-Net architecture is adopted, though the Transformer (Peebles & Xie, 2022) can also be used.

## 2 MULTI-MODAL DIFFUSION MODELS

### 2.1 PRELIMINARIES ON DENOISING DIFFUSION PROBABILITY MODELS (DDPM)

DDPM is a probability generative model that consists of a forward noising process and a reverse denoising process operated on a diffusion space. The forward process gradually adds Gaussian noise into the data, which ultimately become standard Gaussian samples; and the reverse process parameterizes a neural network model to reverse the forward process. Specifically, given a data sample $\mathbf{x}$ from the data distribution, the forward process from time $t-1$ to time $t$ is defined as $q(\mathbf{z}_t \mid \mathbf{z}_{t-1}) = \mathcal{N}(\mathbf{z}_t; \sqrt{1-\beta_t}\,\mathbf{z}_{t-1}, \beta_t\,\mathbf{I})$, where $\mathbf{z}_t$ represents a noisy version of the original data sample $\mathbf{z}_0$ at time $t$; $\{\beta_t\}$ is an increasing sequence converging to 1 (making $\mathbf{x}_t$ converge to a standard Gaussian sample). A reverse process is modeled by a neural network (we consider a U-Net for image data) parameterized by $\boldsymbol{\theta}$ as $p_{\boldsymbol{\theta}}(\mathbf{z}_{t-1} \mid \mathbf{z}_t) = \mathcal{N}(\mathbf{z}_{t-1}; \mu_{\boldsymbol{\theta}}(\mathbf{z}_t, t), \Sigma_{\boldsymbol{\theta}}(\mathbf{z}_t, t))$. Considering all time steps $t = 1, \cdots, T$, the forward and reverse processes define two joint distributions over the same set of random variables $\{\mathbf{z}_0, \cdots, \mathbf{z}_T\}$. By variational principle, a loss corresponding to the evidence lower bound (ELBO) can be derived to optimize the parameterized generative model $\boldsymbol{\theta}$, as $\mathcal{L} = \mathbb{E}_{q(\mathbf{z}_0, \cdots, \mathbf{z}_T)} \left[ -\log p(\mathbf{z}_T) - \sum_{t \geq 1} \log \frac{p_{\boldsymbol{\theta}}(\mathbf{z}_{t-1} \mid \mathbf{z}_t)}{q(\mathbf{z}_t \mid \mathbf{z}_{t-1})} \right]$.

### 2.2 MULTI-MODAL DIFFUSION MODELS

We propose the MT-Diffusion model to jointly model multi-modal data with multi-task learning by generalizing the DDPM framework. We assume each task is associated with one data modality (the data can be the same for different tasks). For example, an unconditional image-generation task is associated with image data, and an image classification task with image-label paired data. Suppose there are $N$ modalities, where modality $i$ is associated with *task data* from space $\mathcal{X}_i$. Let $\mathbf{x}_i$ denote one data sample from the $i$-th modality space, and let $\mathbf{X} \triangleq \{\mathbf{x}_1, \cdots, \mathbf{x}_N\}$ be the union data from the $N$ modalities. We note that our setting is quite general in the sense that the data spaces $\{\mathcal{X}_i\}$ can be heterogeneous, *e.g.*, the image space versus the image-label space as from our previous example.

To deal with potential heterogeneity of modality-data spaces, we propose to define MT-Diffusion in a shared latent space, called *diffusion space* and denoted as $\mathbb{Z}$. To this end, we propose to apply a mapping to project each of the original modality-data space onto the shared diffusion space. We define the mapping with an encoder $E_i$ for task $i$, *i.e.*, $E_i : \mathcal{X}_i \to \mathbb{Z}$, as illustrated in Figure 1. For simplicity, we consider non-parametric or fixed-parameter encoders. The specified encoder designs are detailed in Section 2.2.4. In the following, we first formally define the proposed MT-Diffusion by specifying the forward and reverse processes, as well as deriving the corresponding variaitonal lower bound, by extending the DDPM framework to handle multiple data sources and multi-task losses.

### 2.2.1 FORWARD-REVERSE PROCESSES AND THE VARIATIONAL LOWER BOUND

In our design, the forward and reverse processes will be responsible for integrating multi-modal data information and multi-task losses within the DDPM framework, respectively. This is implemented by first defining joint distributions over data modalities and the diffusion latent variables in both forward and reverse processes. Specif-

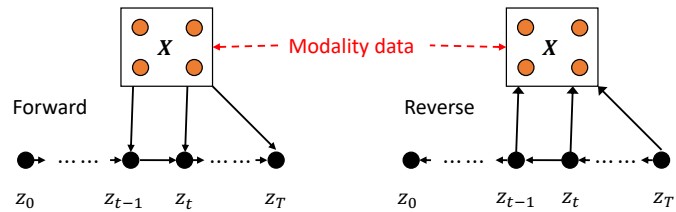

Figure 2: The forward (left) and reverse (right) processes of the proposed MT-Diffusion by jointly modeling a set of task data.

ically, in the forward process, the noising transition from time $t-1$ to time $t$ is defined to be conditioned on the modality data. To this end, we propose to define a joint distribution at time $t$ over the data $\mathbf{X} = \{\mathbf{x}_1, \cdots, \mathbf{x}_N\}$ and the diffusion latent variable $\mathbf{z}_t$, conditioned on information from time $t-1$, to endow the following decomposed form[‡]:

$$q(\mathbf{z}_t, \mathbf{X} \mid \mathbf{z}_{t-1}) = q(\mathbf{z}_t \mid \mathbf{z}_{t-1}, \mathbf{x}_1, \cdots, \mathbf{x}_N) \prod_{i=1}^{N} q_i(\mathbf{x}_i) , \tag{1}$$

where $q(\mathbf{z}_t \mid \mathbf{z}_{t-1}, \mathbf{x}_1, \cdots, \mathbf{x}_N)$ represents the transition distribution of $\mathbf{z}_t$ from time $t-1$ to time $t$, and $\{q_i(\mathbf{x}_i)\}$ denotes prior distributions of the modality data that we assume to be mutually independent for simplicity. We denote this process as *forward aggregation*, and the specific probability distributions will be defined in the next section. Furthermore, the reverse process is defined by simply

---

[‡]We assume modality data are time-independent, although it is also feasible to introduce time dependency.

reversing the forward distributions, resulting in a joint distribution $p_{\boldsymbol{\theta}}(\mathbf{z}_{t-1}, \mathbf{X} \mid \mathbf{z}_t)$ at time $t$, where $\boldsymbol{\theta}$ represents the reverse model parameter. Specifically, starting by sampling $\mathbf{z}_T$ from $p(\mathbf{z}_T)$, we propose to decompose the reverse transition at time $t$ into the following conditional distributions: $p_{\boldsymbol{\theta}}(\mathbf{z}_{t-1}, \mathbf{X} \mid \mathbf{z}_t) = p_{\boldsymbol{\theta}}(\mathbf{z}_{t-1} \mid \mathbf{z}_t) \prod_{i=1}^{N} p_{\boldsymbol{\theta}}(\mathbf{x}_i \mid \mathbf{z}_t)$. The random variable dependency and the general forward-reverse processes are illustrated in Figure 2. Before specifying these distributions, we first derive an objective by matching the joint distributions of the forward and reverse processes. This results in a multi-task ELBO for the proposed MT-Diffusion, based on which a final loss can be defined in Section 2.2.5.

**Theorem 1.** *The negative ELBO of MT-Diffusion endows:* $\mathcal{L} = \mathbb{E}_q [\mathcal{L}_0 + \mathcal{L}_1 + \mathcal{L}_2 + \mathcal{L}_3]$, *where*

$$\mathcal{L}_0 \triangleq KL\left(q(\mathbf{z}_T \mid \mathbf{z}_0, \mathbf{X}) \| p(\mathbf{z}_T)\right), \qquad \mathcal{L}_1 \triangleq \sum_{t>1} KL\left(q(\mathbf{z}_{t-1} \mid \mathbf{z}_0, \mathbf{z}_t, \mathbf{X}) \| p_{\boldsymbol{\theta}}(\mathbf{z}_{t-1} \mid \mathbf{z}_t)\right), \quad (2)$$

$$\mathcal{L}_2 \triangleq \sum_{t \geq 1} \sum_{i=1}^{N} KL\left(q_i(\mathbf{x}_i) \| p_{\boldsymbol{\theta}}(\mathbf{x}_i \mid \mathbf{z}_t)\right), \quad \mathcal{L}_3 \triangleq \log p_{\boldsymbol{\theta}}(\mathbf{z}_0 \mid \mathbf{z}_1).$$

**Remark 1.** *We can see that the prior multi-modal data distributions are within the loss term $\mathcal{L}_2$. If only a single generation task is considered, the sub-loss $\mathcal{L}_2$ will disappeared, reducing to the standard DDPM loss. Our multi-modal diffusion objective defines the posterior of the transition probability $q(\mathbf{z}_{t-1} \mid \mathbf{z}_t, \mathbf{X})$ by conditioning on all modality data (in $\mathcal{L}_1$), and additionally, as formulated in $\mathcal{L}_2$, parameterizes the reverse process to regularize the predicted modality-data distribution $p_{\boldsymbol{\theta}}(\mathbf{x}_i \mid \mathbf{z}_{t-1})$ so that it matches the prior modality data distribution $q_i(\mathbf{x}_i)$.*

### 2.2.2 Forward Aggregation

The forward aggregation mainly deals with the posterior transition probability $q(\mathbf{z}_{t-1} \mid \mathbf{z}_0, \mathbf{z}_t, \mathbf{X})$ in $\mathcal{L}_1$ of equation 2. To derive an explicit form, we start by specifying the forward transition probability $q(\mathbf{z}_t \mid \mathbf{z}_{t-1}, \mathbf{X})$, which can consequently induce the marginal distribution $q(\mathbf{z}_t \mid \mathbf{z}_0, \mathbf{X})$ as well as the posterior transition probability. To integrate different task information, we define the forward transition distribution as a Gaussian distribution by aggregating the task information into the mean parameter. Specifically, we define

$$q(\mathbf{z}_t \mid \mathbf{z}_{t-1}, \mathbf{X}) = \mathcal{N}\left(\mathbf{z}_t; \sqrt{\alpha_t'} \frac{\mathbf{z}_{t-1} + \sum_{i=1}^{N} w_t^{(i)} E_i(\mathbf{x}_i)}{N+1}, (1 - \frac{\alpha_t'}{N+1}) \mathbf{I}\right), \quad (3)$$

where $w_t^{(i)}$ denotes the weight for the $i$-th modality representation at time $t$, and $\{\alpha_t'\}$ are weights to scale the mean and covariance of the Gaussian transition similar to DDPM. By a change of notation $\alpha_t \triangleq \alpha_t'/(N+1)$, the transition distribution can be re-written as $q(\mathbf{z}_t \mid \mathbf{z}_{t-1}, \mathbf{X}) = \mathcal{N}\left(\mathbf{z}_t; \sqrt{\alpha_t}(\mathbf{z}_{t-1} + w_t E(\mathbf{x})), (1 - \alpha_t) \mathbf{I}\right)$, which we will use in the following derivations and implementation. With these transition distributions, multi-task information can be seamlessly incorporated into the diffusion process, which can effectively translate to the reverse process with a parametric model to be defined in Section 2.2.3. Now we can derive the marginal and posterior transition distributions, which turn out to also endow simple forms of Gaussian distributions, stated in Theorem 2.

**Theorem 2.** *Given the transition distribution equation 3, the marginal transition distribution follows*

$$q(\mathbf{z}_t \mid \mathbf{z}_0, \mathbf{X}) = \mathcal{N}\left(\mathbf{z}_t; \sqrt{\bar{\alpha}_t} \mathbf{z}_0 + \sum_{i=1}^{N} \tilde{\alpha}_t^{(i)} E_i(\mathbf{x}_i), (1 - \bar{\alpha}_t) \mathbf{I}\right), \quad (4)$$

*where $\bar{\alpha}_t \triangleq \prod_{i=1}^{t} \alpha_i$, and $\tilde{\alpha}_t^{(i)}$ is recursively defined as $\tilde{\alpha}_t^{(i)} = \sqrt{\alpha_t}\left(w_t^{(i)} + \tilde{\alpha}_{t-1}^{(i)}\right)$ with $\tilde{\alpha}_0^{(i)} \triangleq 0$.*

*Furthermore, the posterior transition follows $q(\mathbf{z}_{t-1} \mid \mathbf{z}_0, \mathbf{z}_t, \mathbf{X}) = \mathcal{N}\left(\mathbf{z}_{t-1}; \tilde{\boldsymbol{\mu}}_t(\mathbf{z}_t, \mathbf{X}), \tilde{\beta}_t \mathbf{I}\right)$, where*

$$\tilde{\boldsymbol{\mu}}_t(\mathbf{z}_t, \mathbf{X}) = \frac{\sqrt{\alpha_t}(1 - \bar{\alpha}_{t-1}) \mathbf{z}_t + (1 - \alpha_t)\sqrt{\bar{\alpha}_{t-1}} \mathbf{z}_0 + \sum_{i=1}^{N} \left(\frac{(1 - \alpha_t)\tilde{\alpha}_t^{(i)}}{\sqrt{\alpha_t}} - (1 - \bar{\alpha}_t) w_t\right) E_i(\mathbf{x}_i)}{1 - \bar{\alpha}_t}$$

$$= \frac{1}{\sqrt{\alpha_t}}\left(\mathbf{z}_t - \frac{1 - \alpha_t}{\sqrt{1 - \bar{\alpha}_t}}\epsilon\right) - \sum_{i=1}^{N} w_t^{(i)} E_i(\mathbf{x}_i), \text{ and } \tilde{\beta}_t = \frac{(1 - \alpha_t)(1 - \bar{\alpha}_{t-1})}{1 - \bar{\alpha}_t}. \quad (5)$$

**Remark 2.** *The posterior transition distribution equation 5 shares a similar form as that in DDPM, with an extra term of $\sum_{i=1}^{N} w_t^{(i)} E_i(\mathbf{x}_i)$ representing information aggregated from all tasks (thus aggregation). Note the aggregation is defined in the forward process, enabling a closed-form posterior but without losing too much modeling expressiveness compared to other complex aggregations.*

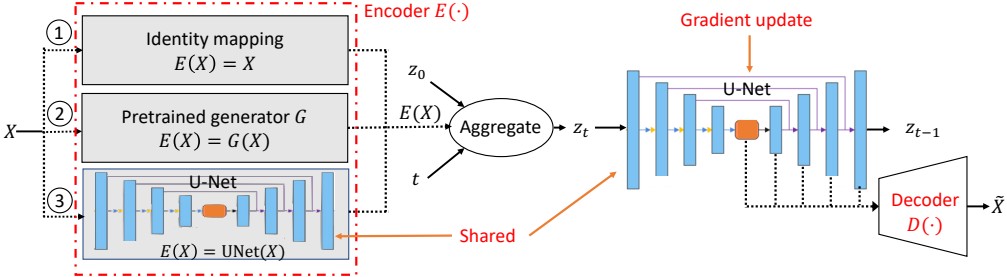

Figure 3: Training pipeline and encoder-decoder design choices. ①②③ indicate three possible choices for the encoder $E(\cdot)$; gray shaped boxes indicate stop gradients; and black dash lines mean possible connections to the encoder and decoder. "Aggregate" is implemented through equation 4.

### 2.2.3 REVERSE PARAMETRIZATION

Based on the ELBO in Theorem 1, the reverse model is responsible for defining two sets of distributions: $p_{\boldsymbol{\theta}}(\mathbf{z}_{t-1} \,|\, \mathbf{z}_t)$ and $p_{\boldsymbol{\theta}}(\mathbf{x}_i \,|\, \mathbf{z}_t)$. The first distribution is similar to that in DDPM, and the second one is induced from decoding the diffusion latent code back to modality-data spaces. To leverage these distributions within a unified architecture for task information sharing, we propose to parameterized the reverse model with a shared backbone network followed by $N$ extra *modality heads*, each corresponding to one modality. The basic structure is illustrated in Figure 1. Specifically, for $p_{\boldsymbol{\theta}}(\mathbf{z}_{t-1} \,|\, \mathbf{z}_t)$, we follow DDPM to define it as Gaussian distributions with mean and covariance denoted as $\mu_{\boldsymbol{\theta}}(\mathbf{z}_t, \mathbf{X})$ and $\sigma_t^2 \mathbf{I}$, respectively. Consequently, the KL-divergence in $\mathcal{L}_1$ of equation 2 reduces to matching the mean of the two Gaussians with a proper weighting scheme depending on $t$. Based on the form of the mean of $q(\mathbf{z}_{t-1} \,|\, \mathbf{z}_0, \mathbf{z}_t, \mathbf{X})$ in equation 5, instead of parameterizing the mean of $p_{\boldsymbol{\theta}}(\mathbf{z}_{t-1} \,|\, \mathbf{z}_t)$, we follow DDPM to parameterize the U-Net to predict the intrinsic noise in $\mathbf{z}_t$ (denoted as $\epsilon$). Specifically, the parametrized U-Net model $\epsilon_{\boldsymbol{\theta}}(\mathbf{z}_t, t)$ is formulated as: $\epsilon_{\boldsymbol{\theta}}(\mathbf{z}_t, t) = \epsilon_{\boldsymbol{\theta}}(\sqrt{\bar{\alpha}_t} \, \mathbf{z}_0 + \tilde{\alpha}_t \, \mathbf{X} + \sqrt{1 - \bar{\alpha}_t} \epsilon, t) \approx \epsilon$.

For the decoding distributions $p_{\boldsymbol{\theta}}(\mathbf{x}_i \,|\, \mathbf{z}_t)$'s in the $\mathcal{L}_2$ term of equation 2, the distribution forms are modality specific. We consider the following two cases in our experiments:

- When modality data are represented as probability vectors, *e.g.*, labels in the classification task, we define $p_{\boldsymbol{\theta}}(\mathbf{x}_i \,|\, \mathbf{z}_t)$ as a discrete distribution, parameterized by the output of the modality head. Consequently, the KL-term in $\mathcal{L}_2$ is equivalent to the cross-entropy loss.

- When modality data are in the form of continuous values, we define $p_{\boldsymbol{\theta}}(\mathbf{x}_i \,|\, \mathbf{z}_t)$ as a Gaussian distribution with the mean parameterized by the output of the modality head. In this case, the KL-divergence in $\mathcal{L}_2$ reduces to the MSE loss, similar to the case for $p_{\boldsymbol{\theta}}(\mathbf{z}_{t-1} \,|\, \mathbf{z}_t)$.

It is worth noting that different from $p_{\boldsymbol{\theta}}(\mathbf{z}_{t-1} \,|\, \mathbf{z}_t)$, the variables $\mathbf{x}_i$ and $\mathbf{z}_t$ in $p_{\boldsymbol{\theta}}(\mathbf{x}_i \,|\, \mathbf{z}_t)$ can be in different feature spaces. Thus, a decoder $D_i(\cdot)$ in the form of one modality head specified above is applied to project the latent code $\mathbf{z}_t$ back to the modality-data space, based on which a proper $p_{\boldsymbol{\theta}}(\mathbf{x}_i \,|\, \mathbf{z}_t)$ is defined, as illustrated in Figure 1. Specifically, the decoding process can be written as:

$$\text{At time } t : \mathbf{z}_t \xrightarrow[\text{denoising}]{\text{Diffusion}} \mathbf{c}_t \triangleq \overline{\text{U-Net}}(\mathbf{z}_t, t; \boldsymbol{\theta}) \xrightarrow[\text{decoding}]{\text{Task } i} \tilde{\mathbf{x}}_i \triangleq D_i(\mathbf{c}_t; \boldsymbol{\theta}) \approx \mathbf{x}_i \ ,$$

where we use "$\overline{\text{U-Net}}$" to denote the output from one particular component of the U-Net, serving as the input to the decoder head (see Section 2.2.4 for more details). In other words, the reverse parameterized model consists of two parts: $\epsilon_{\boldsymbol{\theta}}(\mathbf{z}_t, \mathbf{x}, t)$ and $D_i(\mathbf{z}_t, t; \boldsymbol{\theta})$. Detailed structure designs to integrate the decoders (together with the encoder $E(\cdot)$ in the forward process) into the shared U-Net backbone is discussed in the next section.

### 2.2.4 ENCODER-DECODER DESIGNS

The encoders aim to map different task-data onto the diffusion space, and the decoders project the diffusion latent code from the shared U-Net backbone back to the task-data spaces. As the encoders are associated with the forward process, we propose to avoid introducing extra trainable parameters in the encoders for simplicity. Furthermore, we propose to introduce trainable parameters to the decoders as they are parts of the parameterized reverse model. There are many possible design

choices for the encoders and decoders. *Our guideline is to choose architectures to reuse existing components or some pretrained models as best as possible.* Based on this principle, we recommend the following designs, with the detailed training pipeline and architectures illustrated in Figure 3.

**Encoder Design** We consider three scenarios, indicated by ①②③ in Figure 3: 1) A modality-data space is the same as the diffusion space, *e.g.*, both image spaces. In this case, we can define the encoder as a simple mapping such as the identity mapping ① in Figure 3; 2) A modality-data space is inhomogeneous with the diffusion space, *e.g.*, a label space vs. an image space. In this case, we propose to use either a pretrained generator (② in Figure 3) or the shared U-Net backbone (③ in Figure 3) to transfer modality-data information to the diffusion space. Particularly, for choice ③, we use the cross attention mechanism in the U-Net architecture to map modality-data information to the diffusion space.

**Decoder Design** The decoders are modality and task specific. They accept outputs from one of the U-Net blocks (indicated by black-dash-line connections in Figure 3) and learn to generate the original modality data. For example, in a classification task, the decoder is designed as a classifier that outputs a class label, associated with a cross-entropy loss.

### 2.2.5 TRAINING AND INFERENCE

**Training** We propose a simple training loss for our MT-Diffusion, based on the ELBO equation 2 and the specific forward and reverse parameterization described above. In the ELBO, $\mathcal{L}_0$ is independent of the model parameter $\boldsymbol{\theta}$, thus it can be omitted in training. By substituting the specific distributions into the ELBO and adopting the *simple loss* idea in DDPM (Ho et al., 2020) that ignores the weights for different timesteps, the ELBO equation 2 reduces to the following training loss for the proposed MT-Diffusion:

$$\mathcal{L} \triangleq \tilde{\mathcal{L}}_{\mathrm{mse}} + \lambda \sum_{t \geq 1} \sum_{i=1}^{N} \mathsf{KL}\left(q_i(\mathbf{x}_i) \| p_{\boldsymbol{\theta}}(\mathbf{x}_i \,|\, \mathbf{z}_t)\right) , \text{ where } \mathbf{z}_t \sim q(\mathbf{z}_t \,|\, \mathbf{z}_0, \mathbf{X}) , \tag{6}$$

and $\tilde{\mathcal{L}}_{\mathrm{mse}} \triangleq \mathbb{E}_q \left[ \sum_{t>1} \|\epsilon_{\boldsymbol{\theta}}(\mathbf{z}_t, t) - \epsilon\|^2 + \log p_{\boldsymbol{\theta}}(\mathbf{z}_0 \,|\, \mathbf{z}_1) \right]$ has the same form as the simple loss in DDPM; $\lambda$ is the weight scalar (we set it to 0.1 in our experiments). In particular, the KL terms above can endow closed forms depending on the modality-specific $q_i(\mathbf{x}_i)$. For example, in a classification task with $\mathbf{x}_i$ representing labels, both $q_i(\mathbf{x}_i)$ and $p_{\boldsymbol{\theta}}(\mathbf{x}_i \,|\, \mathbf{z}_t)$ are defined as discrete distributions, making the KL divergence equivalent to the cross entropy. We apply stochastic optimization for model learning. At each iteration, a random timestep $t$ is first sampled. Then the corresponding $\mathbf{z}_t$ is sampled from the forward process with task information aggregation. We then feed $\mathbf{z}_t$ to the reverse model to predict the forward noise and the modality data. Finally, gradient descent is applied to update the model parameter based on the loss equation 6.

**Inference** A distinction of our model is its ability to simultaneously generating multi-modal data. We propose a generic inference procedure that can achieve both unconditional generation (with initially all missing modality data $\mathbf{X}$) or conditional generation (with initially parts of $\mathbf{X}$ known). The basic idea is to estimate the potentially missing modality data from the corresponding heads of the reverse model outputs. The specific algorithm is summarized in Algorithm 1 in Appendix C.

## 3 RELATED WORK

**Diffusion-based Models** Diffusion models (Sohl-Dickstein et al., 2015; Ho et al., 2020; Song et al., 2021) have been state-of-the-art generative models on a variety of applications including image syntheses (Dhariwal & Nichol, 2021), text-to-image generation (Ramesh et al., 2021; Saharia et al., 2022; Yu et al., 2022; Rombach et al., 2022), audio generation (Kong et al., 2021; Liu et al., 2023), video generation (Ho et al., 2022; Harvey et al., 2022; Singer et al., 2023) and text generation (Austin et al., 2021; Li et al., 2022b; He et al., 2022; Gong et al., 2023), *etc*. All these models, however, only focus on a single generation task, in contrast to our multi-task generation.

**Manipulating Diffusion Latent Spaces** There have been significant efforts to manipulating latent spaces of pretrained diffusion models for a variety of downstream tasks, including text-driven image editing, inpainting, completion and *etc* (Gal et al., 2022; Ruiz et al., 2022b; Cohen et al., 2022; Kawar et al., 2022; Meng et al., 2021b; Bau et al., 2021; Avrahami et al., 2022b;a; Bar-Tal et al., 2022; Lugmayr et al., 2022). There are also related works to learning a more discriminative latent space (Zhang et al., 2022; Preechakul et al., 2022) or manipulate a latent space for better representations (Kwon et al., 2023). These methods, however, are task-driven and do not provide

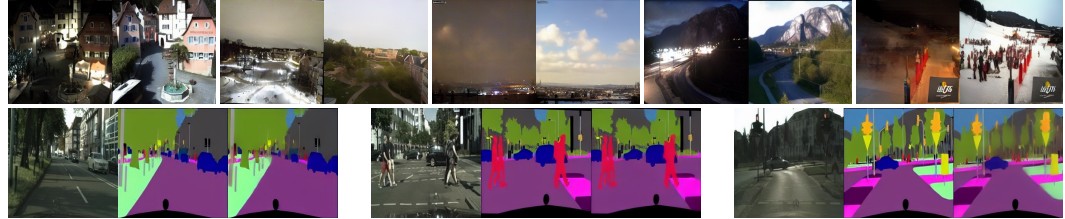

Figure 4: Random samples on the night2day (top, unconditional generation) and cityscape (bottom, conditional generation) datasets. The 3 pictures in each block of the cityscape dataset (bottom) correspond to the conditional image (source), the ground-truth and the inferred target image, respectively.

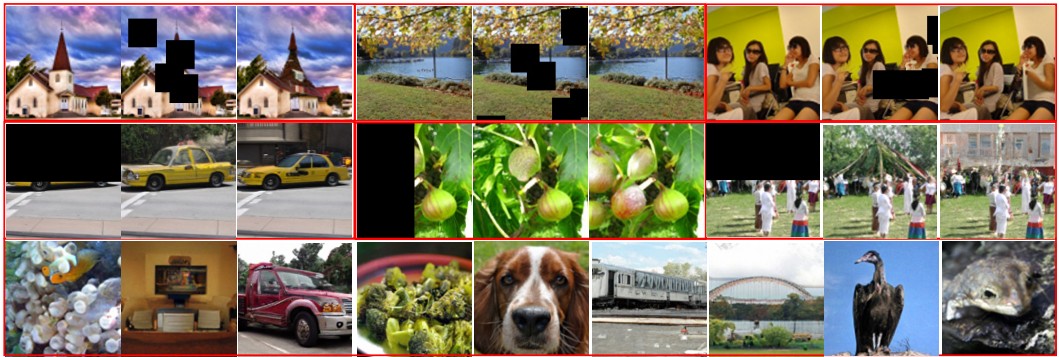

Figure 5: Randomly generated examples of MT-Diffusion with masked-image training. **First row**: image restoration from random masks; images in each block: original, masked and restored images. **Second row**: image restoration from half masking; each block contains two restored images to illustrate generation variance. **Third row**: image generation from scratch with complete masks.

theoretical foundation on the working principles; while all the tasks considered in the literature can be accomplished with our MT-Diffusion framework by incorporating the corresponding task data.

Our model is also related to guided diffusion, which is discussed in details in Appendix D.

## 4 EXPERIMENTS

As a first work on multi-modal diffusion models, we focus on evaluating our MT-Diffusion on the 4 multi-task learning settings mentioned in the Introduction, while leaving other more complicated settings such as incorporating more tasks into training as interesting future work. We describe detailed hyperparameter settings, the four tasks and encoder-decoder designs in Appendix E. In the four experiments, MT-Diffusion for *image translation* and *joint image-representation generative modeling* are mostly for illustration purposes. Thus, the results are mostly qualitative, *e.g.*, Figure 4 demonstrates some visualizations of image-translate. More details are deferred to Appendix E.

### 4.1 MT-DIFFUSION FOR MASKED-IMAGE TRAINING

**Task description and encoder-decoder design** We propose the masked-image training, a new training strategy we design to improve image generation with MT-Diffusion. The task is motivated from the masked-language pretraining paradigm in

Table 1: LPIPS score for masked image recovery. "Mask-$m$" means masking an image with $m$ patches.

| Model | Mask-5 | Mask-10 | Mask-15 | Mask-20 |
|---|---|---|---|---|
| Clean-Masked | 0.311 | 0.414 | 0.461 | 0.491 |
| SDEdit- Meng et al. (2022) | 0.400 | 0.466 | 0.497 | 0.513 |
| MT-Diffusion | 0.035 | 0.068 | 0.099 | 0.133 |

NLP models such as BERT (Devlin et al., 2019a), which achieves significant success with multi-task training. Specifically, in addition to the standard image generation task that generates images from noise, we define an additional task, called *a random inpainting task*, which learns to recover from randomly masked images. Consequently, the forward process is defined to start from a clean-masked

|  | IS↑ | FID↓ | sFID↓ | Precision↑ | Recall↑ |
|---|---|---|---|---|---|
| ADM (un-cond) | 15.64 | 23.22 | 16.53 | 0.57 | 0.60 |
| ADM (class cond) | 22.69 | 16.35 | 17.49 | 0.59 | 0.62 |
| Generation with Masked-Image Training (Section 4.1) | | | | | |
| MT-Diffusion-U | 23.77 | 26.00 | 25.22 | 0.57 | 0.55 |
| MT-Diffusion-X | 34.53 | 9.85 | 15.78 | 0.68 | 0.64 |
| Generation with Joint Image-Label Modeling (Section 4.2) | | | | | |
| MT-Diffusion-M | 15.66 | 33.92 | 21.63 | 0.54 | 0.55 |
| MT-Diffusion-M* | 26.31 | 13.48 | 16.64 | 0.66 | 0.61 |
| MT-Diffusion-E | 11.86 | 41.00 | 22.37 | 0.48 | 0.45 |
| MT-Diffusion-E* | 24.78 | 11.28 | 16.08 | 0.74 | 0.56 |

Table 2: *Training efficiency comparisons on ImageNet-64.*

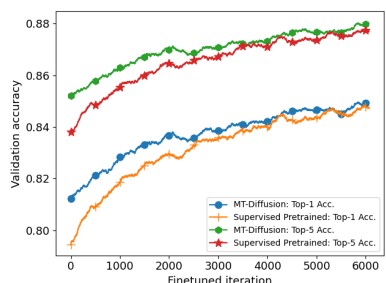

Figure 6: *Classification accuracies vs. supervised finetuning iterations on the ImageNet validation data.*

image pair, and gradually add noise to the pair in the forward process. To create the masked images, we randomly sample $m \sim \mathrm{Uniform}(0 \cdots 10)$ patches of size $16 \times 16$ from the original image and mask them out with zero pixel-values. These patches are placed randomly so they might overlap with each other. We adopt two choices for the noise prediction network $\epsilon_{\theta}(\cdot)$: the first only considers $(\mathbf{z}_t, t)$ as input, denoted as **MT-Diffusion-U**; the other takes $(\mathbf{z}_t, \mathbf{X}, t)$, denoted as **MT-Diffusion-X**. The former is more specifically designed for unconditional generation, whereas the latter is more suitable for *constrained image restoration, a task we defined as inpainting a randomly masked image while keeping the unmasked region unchanged*. Note current state-of-the-art diffusion models do not directly deal with this problem. Existing work for image editing and inpainting such as SDEdit (Meng et al., 2021a) and Dreambooth (Ruiz et al., 2022a) do not explicitly enforce unmask region consistency, thus their inpainting results can change the unmasked region. By considering simultaneously generating the clean-masked image pairs with MT-Diffusion, our model can maximally learn to maintain the consistency of the unmasked regions. Similar to the image-transition experiment, we use the identity map as the encoder, and replicate the output block of the original U-Net as the additional masked-image decoder, which consists of a normalization layer and a convolution layer.

**Results** We implement our method based on the guided diffusion codebase (dif) on ImageNet (Deng et al., 2009). We first demonstrate that our framework can help to improve the training efficiency for image generation. To this end, we compare our and the ADM models (Dhariwal & Nichol, 2021) before convergence at 1M iteration at image resolution 32. For a fair comparison, all models are trained from scratch with exactly the same hyperparameters (thus the numbers are not directly comparable to the reported values). We adopt the same evaluation metrics as the ADM under 5K samples. Quantitative results are shown in Table 2. Note MT-Diffusion-U is designed to generate images from complete random noise while MT-Diffusion-X needs a conditional masked input, which we simply define as a complete mask (all zeros) for this purpose. It is observed both the two

Table 3: Generated image comparisons on ImageNet-128. The subscript "$g$" means classifier guidance; the superscript "*" on ADM indicates results from the released checkpoint; ADM without "*" is the one continued training on the released checkpoint. MT-Diffusion represents the MT-Diffusion-X version; superscript "$f$" indicating continued finetuning on single-task image generation; and the superscript "*" indicates results from the constrained image restoration task with 10% random masking (not directly comparable with the other settings).

|  | IS↑ | FID↓ | sFID↓ | Precision↑ | Recall↑ |
|---|---|---|---|---|---|
| ADM* | 79.95 | 8.46 | 4.92 | 0.67 | 0.66 |
| ADM*$_g$ | 151.10 | 3.56 | 4.63 | 0.79 | 0.57 |
| ADM | 74.04 | 9.62 | 5.61 | 0.66 | 0.64 |
| ADM$_g$ | 140.89 | 4.27 | 5.58 | 0.79 | 0.55 |
| MT-Diffusion | 78.26 | 8.42 | 5.89 | 0.67 | 0.65 |
| MT-Diffusion$_g$ | 81.06 | 8.06 | 5.83 | 0.68 | 0.65 |
| MT-Diffusion$^f$ | 84.22 | 7.01 | 5.99 | 0.69 | 0.64 |
| MT-Diffusion$^f_g$ | 171.65 | 3.51 | 5.73 | 0.83 | 0.53 |
| MT-Diffusion* | 135.35 | 2.15 | 3.86 | 0.72 | 0.68 |
| MT-Diffusion*$_g$ | 138.21 | 2.02 | 3.84 | 0.73 | 0.69 |

variants significantly outperform the unconditioned ADM, indicating the training efficiency and modeling effectiveness of multi-task generative learning via masked-image training. In addition, MT-Diffusion-X is found to perform better than MT-Diffusion-U. We hypothesize this is because the conditional masked-image information makes the training of the model easier and more effective.

In addition to pure image generation, one unique property of MT-Diffusion-X is its ability to simultaneously perform constrained image restoration. To this end, we randomly mask out some testing images with $m = \{5, 10, 15, 20\}$ patches and learn to restore the masked patches. We expect

MT-Diffusion-X to restore masked images without changing unmasked regions. Some example results are illustrated in Figure 5, which clearly demonstrate the strong ability of MT-Diffusion-X for constrained image restoration as well as generation from scratch. For quantitative evaluation, we adopt the LPIPS score (Zhang et al., 2018) that measures the semantic similarity of the original the restored images, and compare our method with one simple baseline from Meng et al. (2022), denoted as SDEdit-. The results are shown in Table 1, where the row of "Clean-Masked" denotes the LPIPS scores of clean-masked image pairs that we include for reference. It is clear that MT-Diffusion-X obtains scores closed to zero, indicating the closed similarity between restored images and original images. SDEdit-, on the other hand, obtains very high LPIPS scores, which are even higher than the Clean-Masked baseline. This is expected since SDEdit- is not specifically designed for such a task.

Finally, to demonstrate the ability of our model to generate high-quality images at convergence, we compare our method with ADM for pure image generation, under a resolution of 128. We train our MT-Diffusion-X from scratch. We evaluate the ADM with two versions: the released checkpoint and a continued trained version from the checkpoint with the same hyperparameters as our model, for a more fair comparison. The results are shown in Table 3 evaluated on 50K samples. Our method achieves comparable performance than ADM (if not better on some metrics such as the IS), while being able to perform more tasks such as the constrained image restoration demonstrated above. It is also noted that the continue-trained version of ADM (started from the released checkpoint) is slightly worse than the released checkpoint, indicating the latter might have been tuned for best performance, and thus it is more fair to compare our methods with the continue-trained version of ADM.

### 4.2 MT-DIFFUSION FOR JOINT IMAGE-LABEL GENERATION MODELING

**Task description and encoder-decoder design**   This is a more heterogeneous case as labels and images are in different spaces. We follow our design principle to use the diffusion U-Net as the encoder for labels via cross attention in the U-Net. The label decoder is an additional head from some layer of the U-Net. We consider two designs: 1) Add one additional MLP layer out of the middle block of the UNet to map the diffusion latent space onto the label space. This introduces minimal extra parameters into the original reverse model but might enforce some discriminative information not helpful for pure image generation. We denote this variant as **MT-Diffusion-M**. 2) Add a pre-defined classifier at the end of the U-Net output. Since the U-Net output can be used to reconstruct the original image, this structure essentially makes the learning of image generation and classification in a sequential manner. In the experiments, we use the classifier provided in the guided diffusion codebase (dif) as the label decoder. We denote this variant as **MT-Diffusion-E**.

**Results**   We adopt the same experiment setting as the previous section. In addition to measuring the generated image quality, we also measure the classification performance using the classifier in MT-Diffusion-E. We find that for such a heterogeneous setting, continuing finetuning both the generator and classifier with single generation and classification tasks, respectively, can significantly improve single-task performance. Consequently, we adopt similar idea of classifier-free guidance to simultaneously learn a pure-generation model along with the multi-task training with the shared U-Net. We denote the finetuned models as **MT-Diffusion-M**$^*$ and **MT-Diffusion-E**$^*$. The results are reported in Table 2. It is observed that MT-Diffusion-M performs better than MT-Diffusion-E, which slightly under-perform the single-task ADM. This is expected and indicates that learning to generate heterogeneous data can trade off single-task performance. We also believe the performance gap is partly due to the un-tuned sub-optimal hyperparameter setting. With single-task finetuning, we can see that both variants outperform ADM in all metrics. To continue finetuning the classifier, we use the training and evaluation script from the codebase (dif), and compare with the pretrained classifier in classifier-guidance ADM (dif). We continue finetuing the classifier from the released checkpoint as a baseline. Top-1 and top-5 accuracies are plotted in Figure 6. It is observed that our classifier consistently outperforms the baseline, although the gap turns smaller with increasing finetuning steps.

## 5 CONCLUSION

We propose the multi-modal diffusion model with multi-task training, a generalization of the standard diffusion model for multi-modal generation. Our model is general and flexible, which can incorporate potentially heterogeneous modality information into a unified diffusion model, compared to training on a single-task setting. We define several multi-task generative problems and test them on our proposed MT-Diffusion. Extensive experiments are performed to verify the effectiveness of our proposed framework. Interesting future works include improving the framework by better network-architectures designs and applying the method to more diverse multi-modal and multi-task settings.

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

## A  ELBO OF THE PROPOSED MULTI-TASK DIFFUSION MODEL: THEOREM 1

We give detailed derivations of our multi-task diffusion ELBO in the following:

$$
\mathcal{L} = \mathbb{E}_q \left[ -\log p(\mathbf{z}_T) - T \sum_{i=1}^{N} \log \frac{1}{q(\mathbf{x}_i)} - \sum_{t \geq 1} \log \frac{p_{\boldsymbol{\theta}}(\mathbf{z}_{t-1} \mid \mathbf{z}_t) \prod_{i=1}^{N} p_{\boldsymbol{\theta}}(\mathbf{x}_i \mid \mathbf{z}_t)}{q(\mathbf{z}_t \mid \mathbf{z}_{t-1}, \mathbf{X})} \right]
$$

$$
= \mathbb{E}_q \left[ -\log p(\mathbf{z}_T) - T \sum_{i=1}^{N} \log \frac{1}{q(\mathbf{x})} - \sum_{t > 1} \log \frac{p_{\boldsymbol{\theta}}(\mathbf{z}_{t-1} \mid \mathbf{z}_t) \prod_{i=1}^{N} p_{\boldsymbol{\theta}}(\mathbf{x}_i \mid \mathbf{z}_t)}{q(\mathbf{z}_{t-1} \mid \mathbf{z}_t, \mathbf{z}_0, \mathbf{X})} \frac{q(\mathbf{z}_{t-1} \mid \mathbf{z}_0, \mathbf{X})}{q(\mathbf{z}_t \mid \mathbf{z}_0, \mathbf{X})} \right.
$$

$$
\left. - \log \frac{p_{\boldsymbol{\theta}}(\mathbf{z}_0 \mid \mathbf{z}_1) \prod_{i=1}^{N} p_{\boldsymbol{\theta}}(\mathbf{x}_i \mid \mathbf{z}_1)}{q(\mathbf{z}_1 \mid \mathbf{z}_0, \mathbf{X})} \right]
$$

$$
= \mathbb{E}_q \left[ -\log \frac{p(\mathbf{z}_T)}{q(\mathbf{z}_T \mid \mathbf{z}_0, \mathbf{X})} - \sum_{t > 1} \log \frac{p_{\boldsymbol{\theta}}(\mathbf{z}_{t-1} \mid \mathbf{z}_t)}{q(\mathbf{z}_{t-1} \mid \mathbf{z}_t, \mathbf{z}_0, \mathbf{X})} \right.
$$

$$
\left. - \sum_{t \geq 1} \sum_{i=1}^{N} \log \frac{p_{\boldsymbol{\theta}}(\mathbf{x}_i \mid \mathbf{z}_t)}{q(\mathbf{x}_i)} - \log p_{\boldsymbol{\theta}}(\mathbf{z}_0 \mid \mathbf{z}_1) \right] \;,
$$

where we use the fact that

$$
q(\mathbf{z}_t \mid \mathbf{z}_{t-1}, \mathbf{x}) = q(\mathbf{z}_t \mid \mathbf{z}_{t-1}, \mathbf{x}, \mathbf{z}_0) = \frac{q(\mathbf{z}_{t-1} \mid \mathbf{z}_t, \mathbf{z}_0, \mathbf{x}) q(\mathbf{z}_t \mid \mathbf{z}_0, \mathbf{x})}{q(\mathbf{z}_{t-1} \mid \mathbf{z}_0, \mathbf{x})}
$$

in the second equality.

## B  CALCULATING THE POSTERIOR DISTRIBUTIONS: THEOREM 2

In our derivation, we will frequently use the following well-known property of Gaussian random variables.

In the following, we first present a Lemma on calculating the posterior distribution of Gaussian random variables, based on which we derive the posterior distribution of our forward process.

**Lemma 3.** *Let* $\epsilon_1 \sim \mathcal{N}(\boldsymbol{\mu}_1, \sigma_1^2 \mathbf{I})$, $\epsilon_2 \sim \mathcal{N}(\boldsymbol{\mu}_2, \sigma_2^2 \mathbf{I})$. *Then, for* $\forall a \geq 0$, $b \geq 0$, *the random variable* $\epsilon \triangleq a\epsilon_1 + b\epsilon_2$ *follows:*

$$
\epsilon \sim \mathcal{N}\left(a\boldsymbol{\mu}_1 + b\boldsymbol{\mu}_2, (a^2\sigma_1^2 + b^2\sigma_2^2)\mathbf{I}\right) \;.
$$

Note in the forward aggregation, $\mathbf{z}_{t-1} + \sum_{i=1}^{N} w_t^{(i)} E_i(\mathbf{x}_i) = \mathbf{z}_{t-1} + w_t^{(1)} \left( \sum_{i=1}^{N} \frac{w_t^{(i)}}{w_t^{(1)}} E_i(\mathbf{x}_i) \right) \triangleq$ $\mathbf{z}_{t-1} + w_t^{(1)} E(\mathbf{X})$, where we define $E(\mathbf{X}) \triangleq \sum_{i=1}^{N} \frac{w_t^{(i)}}{w_t^{(1)}} E_i(\mathbf{x}_i)$. This is equivalent to the form of considering only one extra task with modality-data embedding $E(\mathbf{X})$, *e.g.*, it suffices to only consider two tasks in the proof. Consequently, in the following, we give the derivations of the forward posterior distribution with one additional task, in which case we will drop the task index $i$ in $\mathbf{x}$. Generalizing to $N$ task is straightforward. To derive the marginal distribution $q(\mathbf{z}_t \mid \mathbf{z}_0, \mathbf{x})$, let $\epsilon_t \sim \mathcal{N}(\epsilon; 0, \mathbf{I})$ for $\forall t$. For the forward process, we have

$$
\mathbf{z}_t = \sqrt{\alpha_t}(\mathbf{z}_{t-1} + w_t E(\mathbf{x})) + \sqrt{1 - \alpha_t}\epsilon_t
$$

$$
= \sqrt{\alpha_t}\left( \sqrt{\alpha_{t-1}}(\mathbf{z}_{t-2} + w_{t-1}E(\mathbf{x})) + \sqrt{1 - \alpha_{t-1}}\epsilon_{t-1} + w_t E(\mathbf{x}) \right) + \sqrt{1 - \alpha_t}\epsilon_t
$$

$$
= \sqrt{\alpha_t \alpha_{t-1}}\,\mathbf{z}_{t-2} + \left( w_t \sqrt{\alpha_t} + w_{t-1}\sqrt{\alpha_t \alpha_{t-1}} \right) E(\mathbf{x}) + \sqrt{1 - \alpha_t \alpha_{t-1}}\epsilon
$$

$$
= \sqrt{\alpha_t \alpha_{t-1}}\left( \sqrt{\alpha_{t-2}}(\mathbf{z}_{t-3} + w_{t-2}E(\mathbf{x})) + \sqrt{1 - \alpha_{t-2}}\epsilon_{t-2} \right)
$$

$$
+ \left( w_t \sqrt{\alpha_t} + w_{t-1}\sqrt{\alpha_t \alpha_{t-1}} \right) E(\mathbf{x}) + \sqrt{1 - \alpha_t \alpha_{t-1}}\epsilon
$$

$$
= \sqrt{\alpha_t \alpha_{t-1} \alpha_{t-2}}\,\mathbf{z}_{t-3} + \left( w_t \sqrt{\alpha_t} + w_{t-1}\sqrt{\alpha_t \alpha_{t-1}} + w_{t-2}\sqrt{\alpha_t \alpha_{t-1} \alpha_{t-2}} \right) E(\mathbf{x})
$$

$$
+ \sqrt{1 - \alpha_t \alpha_{t-1} \alpha_{t-2}}\epsilon \;,
$$

where we apply Lemma 3 in the third equation to consolidate the two random Gaussian variables $\epsilon_{t-1}$ and $\epsilon_t$ into $\epsilon$.

Let $\bar{\mathbf{x}}_t \triangleq \prod_{i=1}^{t} \alpha_i$, $\tilde{\alpha}_t = \sum_{i=1}^{t} \prod_{j=i}^{t} \alpha_j$, and define $\tilde{\alpha}_0 = 0$. We have

$$\tilde{\alpha}_t = \sqrt{\alpha_t} \left( w_t + \tilde{\alpha}_{t-1} \right), \text{ and}$$

$$q(\mathbf{z}_t \mid \mathbf{z}_0, \mathbf{x}) = \mathcal{N}\left( \mathbf{z}_t; \sqrt{\bar{\alpha}_t}\, \mathbf{z}_0 + \tilde{\alpha}_t E(\mathbf{x}), (1 - \bar{\alpha}_t)\, \mathbf{I} \right) . \tag{7}$$

As a special case, if we define the forward transition distribution by letting $w_t = 1$, we will have:

$$\tilde{\alpha}_t = \sqrt{\alpha_t} \left( 1 + \tilde{\alpha}_{t-1} \right) . \tag{8}$$

**Lemma 4** (Murphy (2022)). *Define the following distributions for the prior and likelihood:*

$$p(\mathbf{x}) = \mathcal{N}\left( \mathbf{x}; \boldsymbol{\mu}, \Lambda^{-1} \right), \ \ p(\mathbf{y} \mid \mathbf{x}) = \mathcal{N}\left( \mathbf{y}; \mathbf{A}\,\mathbf{x} + \mathbf{b}, \mathbf{L}^{-1} \right) .$$

*Let $\Sigma = \left( \Lambda + \mathbf{A}^T \mathbf{L}\, \mathbf{A} \right)^{-1}$. Then the posterior follows:*

$$p(\mathbf{x} \mid \mathbf{y}) = \mathcal{N}\left( \mathbf{x}; \Sigma\left( \mathbf{A}^T \mathbf{L}(\mathbf{y} - \mathbf{b}) + \Lambda\boldsymbol{\mu} \right), \Sigma \right) .$$

In our case in equation 7, we have $\boldsymbol{\mu} \triangleq \sqrt{\bar{\alpha}_{t-1}}\, \mathbf{z}_0 + \tilde{\alpha}_{t-1} E(\mathbf{x})$, $\Lambda \triangleq \frac{1}{1 - \bar{\alpha}_{t-1}} \mathbf{I}$, $\mathbf{A} \triangleq \sqrt{\alpha_t}\, \mathbf{I}$, $\mathbf{b} \triangleq \sqrt{\alpha_t} w_t E(\mathbf{x})$, $\mathbf{L} \triangleq \frac{1}{1 - \alpha_t} \mathbf{I}$, and $\Sigma = \frac{(1 - \alpha_t)(1 - \bar{\alpha}_{t-1})}{1 - \bar{\alpha}_t} \mathbf{I}$. Thus, the posterior $q(\mathbf{z}_{t-1} \mid \mathbf{z}_t, \mathbf{z}_0, \mathbf{x}) = \mathcal{N}\left( \mathbf{z}_{t-1}; \tilde{\boldsymbol{\mu}}_t(\mathbf{z}_t, \mathbf{z}_0, \mathbf{x}), \tilde{\beta}_t\, \mathbf{I} \right)$, where

$$\tilde{\boldsymbol{\mu}}_t(\mathbf{z}_t, \mathbf{z}_0, \mathbf{x}) = \frac{\sqrt{\alpha_t}(1 - \bar{\alpha}_{t-1})\, \mathbf{z}_t + (1 - \alpha_t)\sqrt{\bar{\alpha}_{t-1}}\, \mathbf{z}_0 + \left( (1 - \alpha_t)\tilde{\alpha}_{t-1} - \alpha_t(1 - \bar{\alpha}_{t-1})w_t \right) E(\mathbf{x})}{1 - \bar{\alpha}_t}$$

$$\tag{9}$$

$$= \frac{\sqrt{\alpha_t}(1 - \bar{\alpha}_{t-1})\, \mathbf{z}_t + (1 - \alpha_t)\sqrt{\bar{\alpha}_{t-1}}\, \mathbf{z}_0 + \left( (1 - \alpha_t)\tilde{\alpha}_t/\sqrt{\alpha_t} - (1 - \bar{\alpha}_t)w_t \right) E(\mathbf{x})}{1 - \bar{\alpha}_t}$$

$$\tilde{\beta}_t = \frac{(1 - \alpha_t)(1 - \bar{\alpha}_{t-1})}{1 - \bar{\alpha}_t} .$$

From the marginal distribution $q(\mathbf{z}_t \mid \mathbf{z}_0, \mathbf{x})$, we have

$$\mathbf{z}_t = \sqrt{\bar{\alpha}_t}\, \mathbf{z}_0 + \tilde{\alpha}_t E(\mathbf{x}) + \sqrt{1 - \bar{\alpha}_t}\epsilon$$

$$\rightarrow \mathbf{z}_0 = \frac{1}{\sqrt{\bar{\alpha}_t}} \left( \mathbf{z}_t - \tilde{\alpha}_t E(\mathbf{x}) - \sqrt{1 - \bar{\alpha}_t}\epsilon \right) . \tag{10}$$

Substituting equation 10 into equation 9, we have

$$\tilde{\boldsymbol{\mu}}_t(\mathbf{z}_t, \mathbf{x}, t)$$

$$= \frac{\sqrt{\alpha_t}(1 - \bar{\alpha}_{t-1})\, \mathbf{z}_t + (1 - \alpha_t)\sqrt{\bar{\alpha}_{t-1}}\left( \frac{1}{\sqrt{\bar{\alpha}_t}} \left( \mathbf{z}_t - \tilde{\alpha}_t E(\mathbf{x}) - \sqrt{1 - \bar{\alpha}_t}\epsilon \right) \right)}{1 - \bar{\alpha}_t}$$

$$+ \frac{\left( \frac{(1 - \alpha_t)\tilde{\alpha}_t}{\sqrt{\alpha_t}} - (1 - \bar{\alpha}_t)w_t \right) E(\mathbf{x})}{1 - \bar{\alpha}_t}$$

$$= \frac{\left( \sqrt{\alpha_t}(1 - \bar{\alpha}_{t-1}) + (1 - \alpha_t)/\sqrt{\alpha_t} \right)\, \mathbf{z}_t - (1 - \alpha_t)w_t E(\mathbf{x}) - (1 - \alpha_t)\sqrt{\frac{(1 - \bar{\alpha}_t)\tilde{\alpha}_{t-1}}{\bar{\alpha}_t}}\epsilon}{1 - \bar{\alpha}_t}$$

$$= \frac{1}{\sqrt{\alpha_t}} \left( \mathbf{z}_t - \frac{1 - \alpha_t}{\sqrt{1 - \bar{\alpha}_t}}\epsilon \right) - w_t E(\mathbf{x})$$

Thus, similar to the standard DDPM, we can parameterize the backward denoising process with a neural network to predict the added noise $\epsilon$, except that in our case, the neural network $\epsilon_{\boldsymbol{\theta}}$ would take $(\mathbf{z}_t, \mathbf{x}, t)$ as the input, *i.e.*,

$$\epsilon_{\boldsymbol{\theta}}(\mathbf{z}_t, \mathbf{x}, t) = \epsilon_{\boldsymbol{\theta}}(\sqrt{\bar{\alpha}_t}\, \mathbf{z}_0 + \tilde{\alpha}_t\, \mathbf{x} + \sqrt{1 - \bar{\alpha}_t}\epsilon, \mathbf{x}, t) \approx \epsilon .$$

## C  INFERENCE

The inference algorithm is shown in Algorithm 1. If not explictly stated, the number be timesteps is set to $T = 1000$.

---

**Algorithm 1** MT-Diffusion Inference

---

1: $\mathbf{z}_T \sim \mathcal{N}(\mathbf{0}, \mathbf{I})$
2: **if** Modality data $\mathbf{x}_i$ ($\forall i$) not initially available **then**
3:      Randomly initialize modality data $\mathbf{x}_i$
4: **end if**
5: **for** $t = T, \cdots, 1$ **do**
6:      $\epsilon \sim \mathcal{N}(\mathbf{0}, \mathbf{I})$ if $t > 1$, else $\epsilon = \mathbf{0}$
7:      $\mathbf{e}_i = E(\mathbf{x}_i), \forall i$                      ▷ Get modality data encoding
8:      $\epsilon_{\boldsymbol{\theta}}(\mathbf{z}_t, t), \tilde{\mathbf{X}} = \texttt{U-Net}(\mathbf{z}_t, t)$      ▷ Get estimated noise and predicted modality data
9:      **if** $\mathbf{x}_i$ ($\forall i$) not initially available **then**
10:        Update $\mathbf{x}_i$ from the new $\tilde{\mathbf{X}}, \forall i$
11:      **end if**
12:      $\mathbf{z}_{t-1} = \frac{1}{\sqrt{\alpha_t}} \left( \mathbf{z}_t - \frac{1-\alpha_t}{\sqrt{1-\bar{\alpha}_t}} \epsilon_{\boldsymbol{\theta}}(\mathbf{z}_t, \mathbf{x}, t) \right) - \sum_{i=1}^{N} w_t^{(i)} \mathbf{e}_i + \tilde{\beta}_t \epsilon$      ▷ Update diffusion latent
13: **end for**
14: **return** $\mathbf{z}_0, \mathbf{X}$

---

## D  RELATED WORKS

**Connections to Classifier and Classifier-Free Guidance**   Guided diffusion models aim to leverage prior knowledge from various guidance information for better controllable generation. For example, the classifier guidance method uses the gradient of a pretrained classifier to perturb the reverse process to generate from a class-conditional distribution (Dhariwal & Nichol, 2021). The classifier-free guidance simultaneously learns a guidance model using the same generation network of the diffusion model (Ho & Salimans, 2022). The works that try to utilize external data such as the retrieval-augmented based methods (Blattmann et al., 2022; Long et al., 2022) can also be considered as a special type of guidance. Although the final formulation has some connections with our method (see Appendix D), guided diffusion models essentially only handle a single generation task. Our method, on the other hand, can model multiple tasks within a unified diffusion model.

Our MT-Diffusion formulation endows an closed connection with the classifier guidance and classifier-free guidance mechanisms. Specifically, from equation 5, if one defines the encoder $E(\cdot)$ as the gradient from a pretrained classifier, the posterior mean recovers the one for classifier guidance. By contrast, if one defines the encoder with the reverse U-Net, the posterior mean calculation in equation 5 recovers the classifier-free guidance mechanism. However, an important difference is the forward process, where our framework is designed to aggregate information from different encoders for multi-task learning, whereas both classifer guidance and classifer-free guidance do not. Overall, our method constitutes a broader framework that can be applied to different scenarios, including image transition, masked-image pretraining, joint image-label and image-representation generation investigated in the experiments.

**Multi-Task Learning**   Multi-Task Learning (MTL) is a paradigm in machine learning that involves training a model to perform multiple tasks simultaneously, with the idea that knowledge gained from one task can help improve the performance on other related tasks. Recent development has mainly focused on multi-task learning for predictive models instead of generative models. Apart from investigating theory in multi-task learning (Wang et al., 2021; Tiomoko et al., 2021; Tripuraneni et al., 2020; Wu et al., 2020), many existing works explore different techniques to boost model performance with multi-task learning, including but not limited to architecture designs (Heuer et al., 2021; Ruder et al., 2017; Ye & Xu, 2023; Sharma et al., 2023; Chen et al., 2022), optimization algorithms (Senushkin et al., 2023; Fernando et al., 2023; Jiang et al., 2023; Phan et al., 2022) and task relationship learning (Hu et al., 2022; Ilharco et al., 2022). Recent research interest has also be expanded to applying multi-task learning in generative models (Bao et al., 2022; Liu et al., 2018). However, the generative models are limited to more traditional models such as VAE and GAN. And

there is limited work on studying multi-task learning for diffusion models. Our work represents one of the first works on integrating multi-modal generation with multi-task learning in a diffusion model, aiming to further improve generative performance and expand the scope of state-of-the-art diffusion models.

**More Recent Development on Diffusion Models**  There is some recent effort trying to develop multi-task diffusion models. For example, the Versatile Diffusion (Xu et al., 2022) The Versatile diffusion focuses on developing new neural architectures that make different tasks interact with each other within the single-task diffusion framework. This is different from our work in that ours not only introduces a novel neural architecture but also generalizes the single-task diffusion in terms of the loss function. We believe the Versatile diffusion architecture can be incorporated into our framework for more flexible modeling.

Previous efforts have also focused on efficient training of diffusion models, *e.g.*, P2-Weighting (Choi et al., 2022), Min-SNR (Hang et al., 2023), ANT Go et al. (2023), and Task Routing (Park et al., 2023). Our model is orthogonal to these works with no technical overlap. Consequently, we believe there is room to incorporate these techniques into our framework for further improvement, an interesting future direction to be explored.

## E  DETAILED EXPERIMENTAL SETTINGS AND EXTRA RESULTS

In addition to evaluating on some other datasets that will be described in the specific tasks, we mainly rely on the ImageNet-1K dataset (Deng et al., 2009) with resolutions of $64 \times 64$ and $128 \times 128$, where we adopt the pre-defined training and validation splits. All experiments are conducted on a A100 GPU server consists of 8 GPUs, with a batchsize of 64, if not explicitly specified. When evaluating generation quality, we follow and adopt the popular Inception Score (IS), FID score, sFID score, Precision and Recall metrics (dif), calculated on 10K or 50K samples, where the former is for computational efficiency and latter for comparing with existing results. We note that due to our different hyperparameter settings (specified in the Appendix), some of our results are not directly comparable to some reported results in previous works. For fair comparisons, we rerun some of the baselines on our settings that are consistent with our method. One additional hyperparameter of our model is the task weights in equation 3, which we set to $w_t^{(i)} = t/(1000 - t)$ to mitigate some potentially negative influence from some heterogeneous tasks on the generated image quality when $t$ is small. We follow most of the parameter settings as in the codebases.

The training procedure is summarized in Algorithm 2.

---

**Algorithm 2** MT-Diffusion Training

1: **repeat**
2:    $\mathbf{z}_0 \sim q(\mathbf{z}_0)$, $\mathbf{X} \sim q(\mathbf{X})$                           ▷ Sample $\mathbf{z}_0$ and modality data $\mathbf{X}$
3:    $t \sim \text{Scheduler}(1, \cdots, T)$
4:    $\{\mathbf{e}_i\} = E(\mathbf{X})$                                            ▷ Get modality data encoding
5:    $\epsilon \sim N(\mathbf{0}, \mathbf{I})$
6:    $\mathbf{z}_t \sim q(\mathbf{z}_t \,|\, \mathbf{z}_0, \mathbf{X})$                                    ▷ Forward aggregation via equation 4
7:    $\mathbf{e}_i = E(\mathbf{x}_i), \forall i$                                        ▷ Get modality data encoding
8:    $\epsilon_{\boldsymbol{\theta}}(\mathbf{z}_t, t), \tilde{\mathbf{X}} = \text{U-Net}(\mathbf{z}_t, t)$      ▷ Noise and modality data prediction via the reverse model
9:    Take gradient descent step based on the loss equation 6
10: **until** Converged

---

### E.1  EXPERIMENT SETTINGS FOR IMAGE TRANSLATION WITH MT-DIFFUSION

For the Cityscape dataset, the modality data corresponds to the semantic segmentation maps; and for the night2day dataset, the modality data corresponds to images of day time. We adopt the latent diffusion codebase from ldm, and use the provided checkpoints of the VQ-VAE encoder-decoder (kl-f8.pt). We use the VQ-VAE encoder as the encoder for modality data; and construct an additional output head by duplicate the original output block of the U-Net structure as the decoder to generate

the modality data, which consists of a normalization layer, a SiLU layer, and a convolution layer. We use the default hyper-parameters for training the models for the two datasets, summarized as:

- Attention resolutions: (32, 16, 9)
- Diffusion steps: 1000
- Learn sigma: False
- Noise schedule: Linear
- #channels: 320
- #heads: 8
- #res blocks: 2
- Resblock updown: False
- Use scale shift norm: False
- Learning rate: 1.0e-4
- Batch size: 32

**Task description and encoder-decoder design** This is a more modality-homologous setting. We adopt two standard datasets, the Cityscale dataset for semantic-labels to photo translation (Cordts et al., 2016) and the night2day dataset for night-to-day

Table 4: Classification performance on Cityscape dataset.

| Model | Per-pixel acc. | Per-class acc. | Class IOU |
|---|---|---|---|
| CycleGAN Zhu et al. (2017) | 0.58 | 0.22 | 0.16 |
| pix2pix Isola et al. (2017) | 0.85 | 0.40 | 0.32 |
| InternImage-H Wang et al. (2022) | - | - | 0.86 |
| Single-task diffusion | 0.72 | 0.54 | 0.32 |
| MT-Diffusion | 0.95 | 0.85 | 0.70 |

photo translation (Laffont et al., 2014). We adopt the public codebase of latent diffusion model (LDM) (ldm). For the translation problem, the task data (original and translated images) are in the same data space, thus we do not need to explicitly define separate encoders $E_i(\cdot)$ for the modality data. Instead, we use the same pretrained image encoder in LDM to map all images to the diffusion latent space. We add another head at the end of the U-Net as the decoder for target translated images.

**Results** We perform image translation by generating target images conditioned on source images based on Algorithm 1. Some example generated image are illustrated in Figure 4. For quantitative evaluation, we follow Zhu et al. (2017) to measure the performance in terms of per-pixel accuracy, per-class accuracy and class IOU, and compare it with exiting methods (Zhu et al., 2017; Isola et al., 2017; Wang et al., 2022). The results are shown in Table 4. It is cleared that our model obtains the best accuracy compared to the baselines, except for the state-of-the-art InternImage-H model, which is a much larger image foundation model pretrained on web-scaled data, thus it is not comparable. We also calculate the IS and FID scores on the night2day dataset. Note prior work did not typically calculate these scores. We obtain an FID score of 37.93 and an IS of 3.94, and the IS score is even slightly better than that of the ground-truth data (3.65). As a comparison, the FID and IS scores with a single-task diffusion are 40.73 and 3.84, respectively.

## E.2 MT-DIFFUSION FOR MASKED-IMAGE TRAINING

In this task, the modality data is a randomly masked version of the original images. To create a randomly masked image, we random sample a coordinate $(x, y)$ that is within the image, then we masked out a patch $(x : \min(x + 16, 64)), y : \min(y + 16, 64)$ by setting the corresponding pixel values to zeros. We repeat this process for $m$ to control the ratio of masked regions. We adopt the latent diffusion codebase from dif. We simply define the encoder as the identity map, and define the decoder for the masked images by replicating the output block of the original U-Net, similar to the above Image Translation experiment. We adopt the default hyper-parameters for training the models, if not specified below.

- Diffusion steps: 1000
- Rescale learned sigmas: False

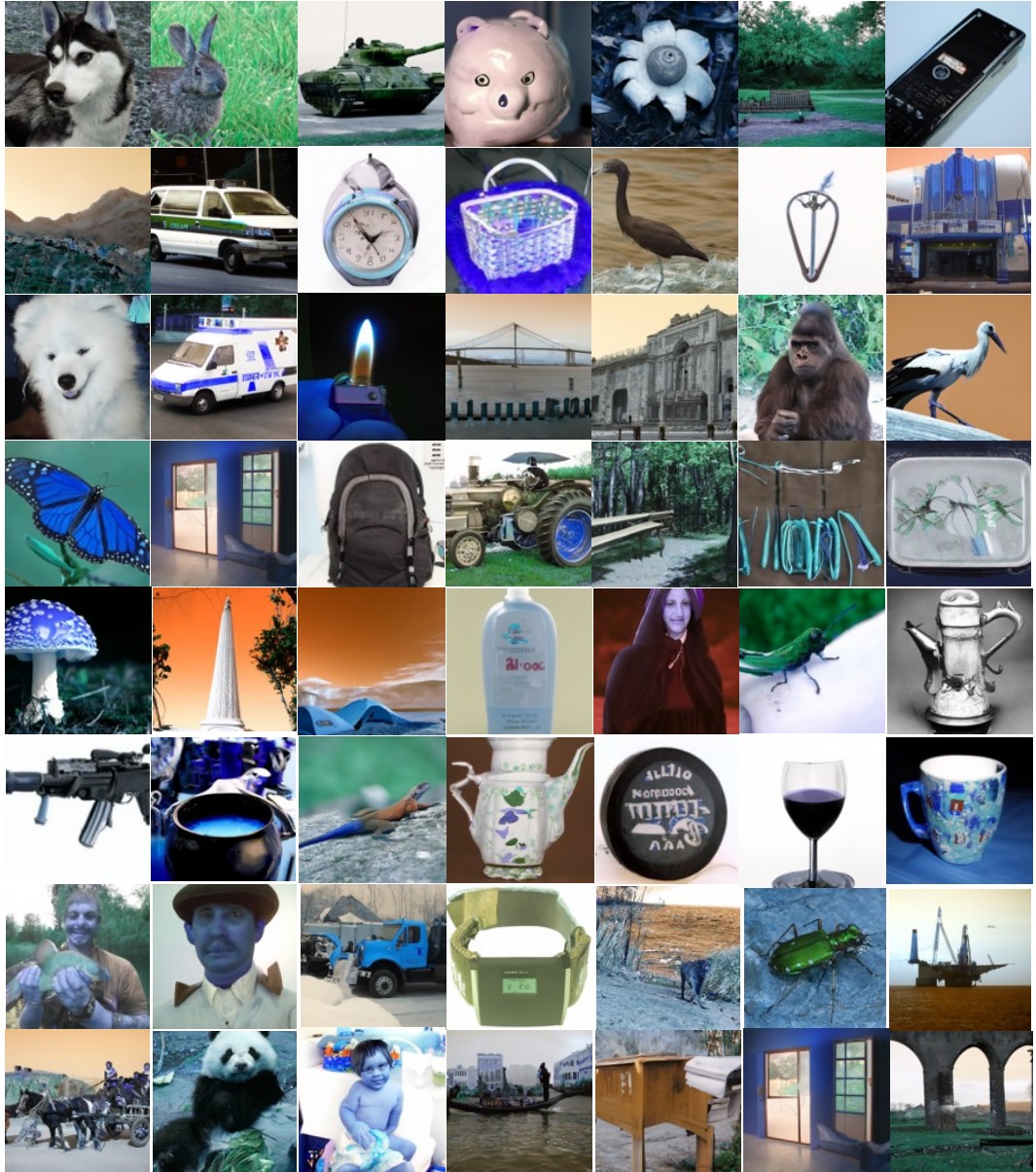

Figure 7: Random samples from scratch with MT-Diffusion by masked-image training on ImageNet-128×128.

- Rescale timesteps: False

- Noise schedule: cosine

- #channels: 192

- #res blocks: 3

- Learning rate: 7.0e-5

- Batch size: 80

More random samples from scratch and image restoration results from both random masking and half masking are illustrated in Figure 7, 8, 9 and 10.

### E.3    MT-Diffusion for Joint Image-Label Generation Modeling

In this task, the modality data are discrete labels. We use the original U-Net as the encoder, which tasks a noisy image, a label and a timestep as input. For simplicity, we set the noisy image and the timestep to be zeros, although we believe better results can be obtained by jointly encoding with such information. For decoders, we proposes two options, one go out of the middle block of the U-Net and the other go out of the output block, as described in the main text. For the one from the middle block, we simply add one fully connected layer to define the decoder; and for the one from the output block, we adopt the pre-defined classifier from the codebase dif as the decoder. We adopt the default hyper-parameters for training the models, if not specified below. First, for MT-Diffusion-M:

- Diffusion steps: 1000
- Rescale learned sigmas: False
- Rescale timesteps: False
- Noise schedule: cosine
- #channels: 192
- #res blocks: 3
- Learning rate: 7.0e-5
- Batch size: 75

The setting for MT-Diffusion-E is the same as MT-Diffusion-M, except with some extra hyper-parameters for the pre-difined classifier:

- Classifier_attention_resolutions: ( 32,16,8)
- Classifier_depth: 4
- Classifier_pool: attention
- Classifier_resblock_updown: True
- Classifier_use_scale_shift_norm: True

### E.4    MT-Diffusion for Joint Image-Representation Generation Modeling

**Task description and encoder-decoder design**    Finally, we apply MT-Diffusion for joint image-representation generation. The setting is similar to the image-label generation setting in Section 4.2, by replacing the label data with image representation from the CLIP model Radford et al. (2021). Similarly, we use the original U-Net as the encoder for image representations via the cross-attention mechanism. For the decoder, we append a two-layer MLP to the output of the middle block of the U-Net, which is expected to output image representations. The MLP project the tensor from middle block to dimension of 1024, followed by a ReLU layer, and finally another layer output tensor of 1024. For MT-Diffusion for Joint Image-Representation:

- Diffusion steps: 1000
- Learning rate: 1.0e-5
- Batch size: 2048

**Results**    We conduct large-scale experiments based on the pretrained stable diffusion model Rombach et al. (2022)[§], by continuing finetuning the model on the LAION dataset Schuhmann et al. (2022) with our MT-Diffusion. We adopt the default hyperparameter setting as that in the codebase. Due to the large-scale nature, it is challenging to make fair quantitative comparisons with related methods. Thus, we only show some generated examples from our method, and leave more extensive comparisons as future work. Some randomly generated examples are shown in Figure 11 and 12, demonstrating impressive generated quality results.

We also provide a visulized comparison between our MT-diffusion with the stable diffusion baseline in Figure 13 and 14. From the generated images, it appears that our method can understand the semantic meaning of the images and generate better looking images.

---

[§]https://huggingface.co/stabilityai/stable-diffusion-2

### E.5 DISCUSSION

**Computation Efficiency**    Our multi-modal setting only adds small modality heads to decode back to the modality space. In our two-modality setting, the additional computational overhead is minimal, amounting to approximately 10% more training time per iteration than a pure single-task diffusion model. Importantly, our model remains significantly more efficient than training two individual diffusion models for the two modalities separately in terms of both time and storage efficiency.

**Extra Experiments**    During the rebuttal, we try to design new experiments to demonstrate 1) our model is better than a pure condition model on two modalities; 2) negative transfer phenomenon in our model.

For 1), we compare our model with a pure condition model that learns to recover images from random masked images. We run the experiments on the small CIFAR-10 dataset. We observe that our model can converge faster than the pure conditional baseline, while both converge to comparable final results in terms of both IS and FID scores. However, we would like to emphasize that our model not only can do conditional generation, but also joint generation for multiple modals. Thus, our model represents a more flexible generative model framework.

For 2), we plan to test our model on more tasks and modallities. Specifically, we plan to train our model on 5 tasks to simultaneously learning to generate original images,, masked images, corresponding captions, random captions, and class labels. After spending significant efforts in implementation, we find it takes too much work to finish the experiment. In addition, we are in lack of GPU resources. Thus, unfortunately, we have to postpone this large-scale experiment. However, we wish to point out that our current results actually have the implications that more similar tasks tend to have more positive transfers. For example, comparing the following two settings indicated in Table 2: 1) simultaneously generating images and the corresponding masked images; 2) simultaneously generating images and the corresponding labels. The former two tasks are considered closer as they are in the same data space. And from Table 2, we can clearly see that the former setting (Generation with Masked-Image Training (Section 4.1)) outperforms the latter one (Generation with Joint Image-Label Modeling (Section 4.2)), indicating there are more positive transfers in the former setting.

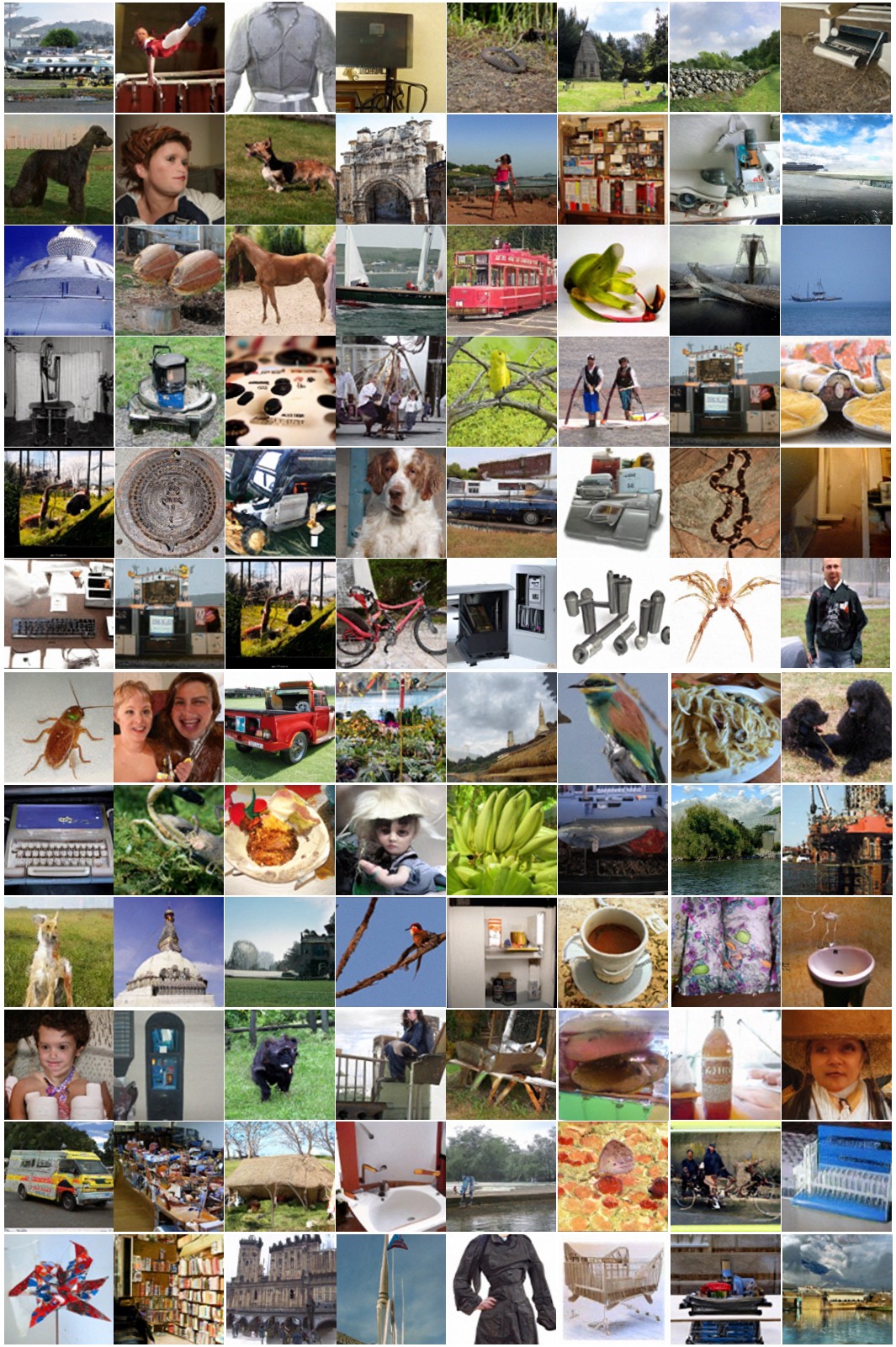

Figure 8: Random samples from scratch with MT-Diffusion by masked-image training on ImageNet-64×64.

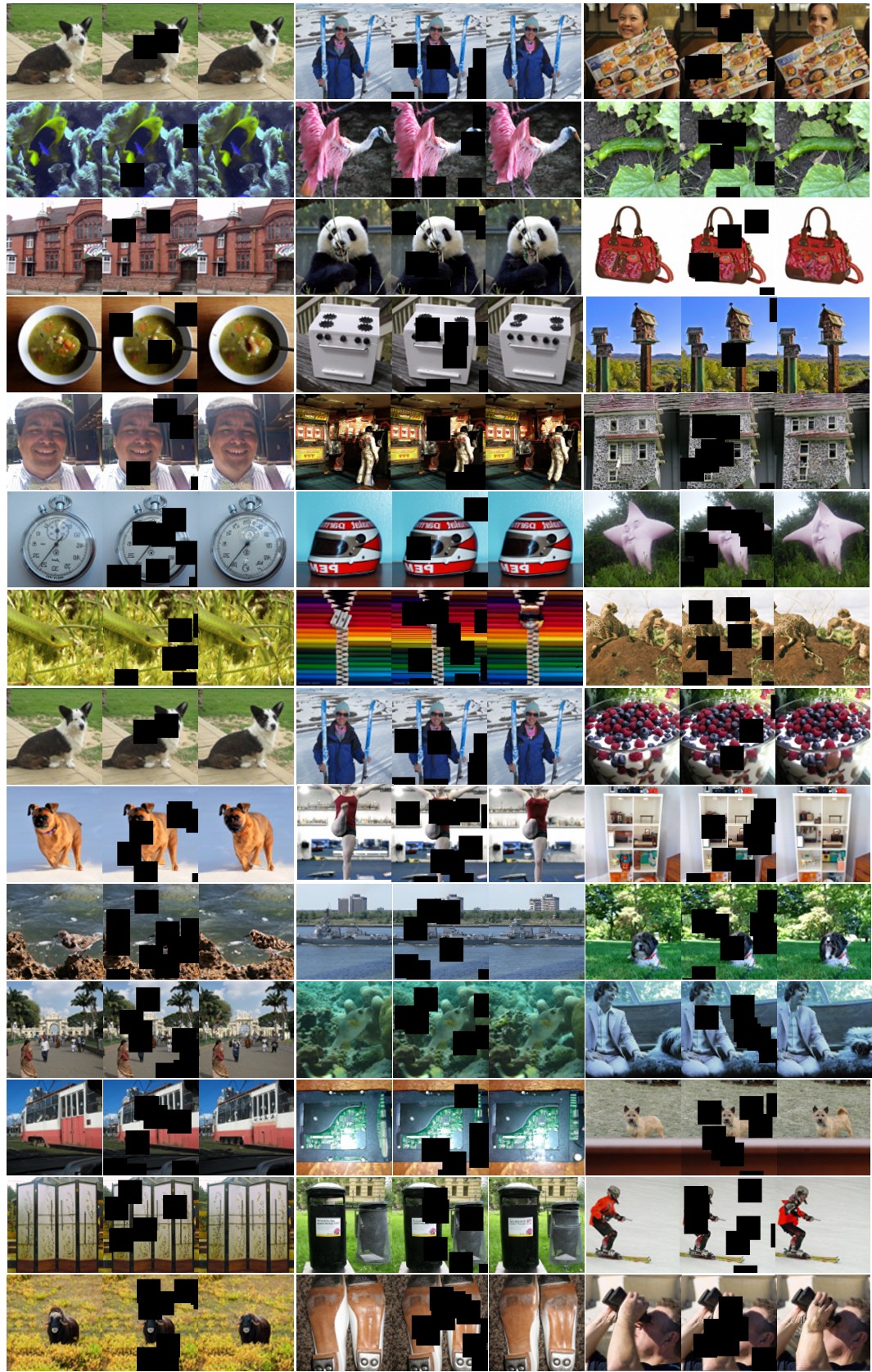

Figure 9: Random samples for image restoration from random masking on ImageNet-64×64. In each block, the three images are original image, masked image and restored image, respectively.

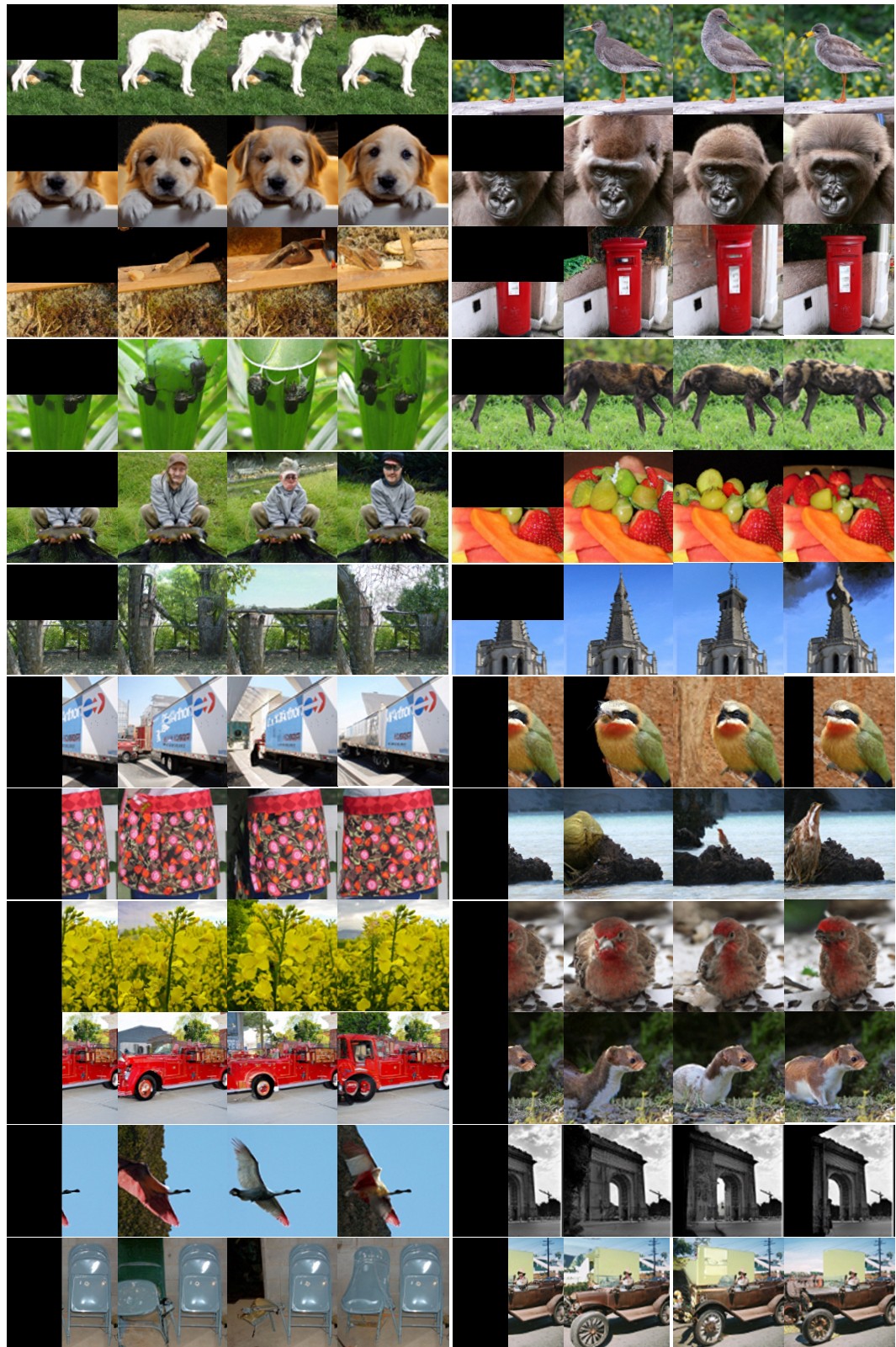

Figure 10: Random samples for image restoration from half masking on ImageNet-64×64. In each block, the first image is the masked image, the rest three are different restored imaegs.

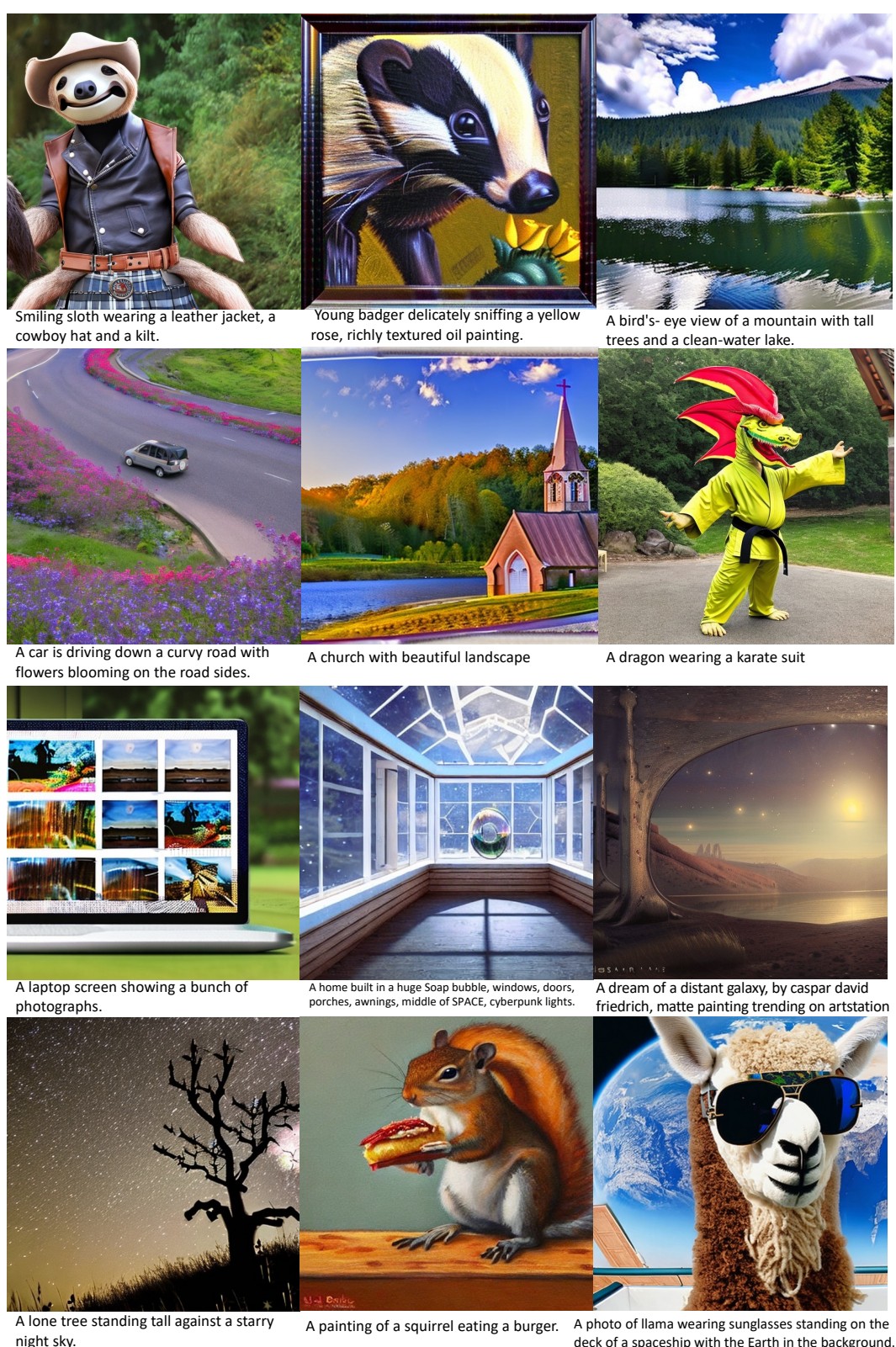

Figure 11: Random samples for text-to-image generation finetuned on stable diffusion v2.

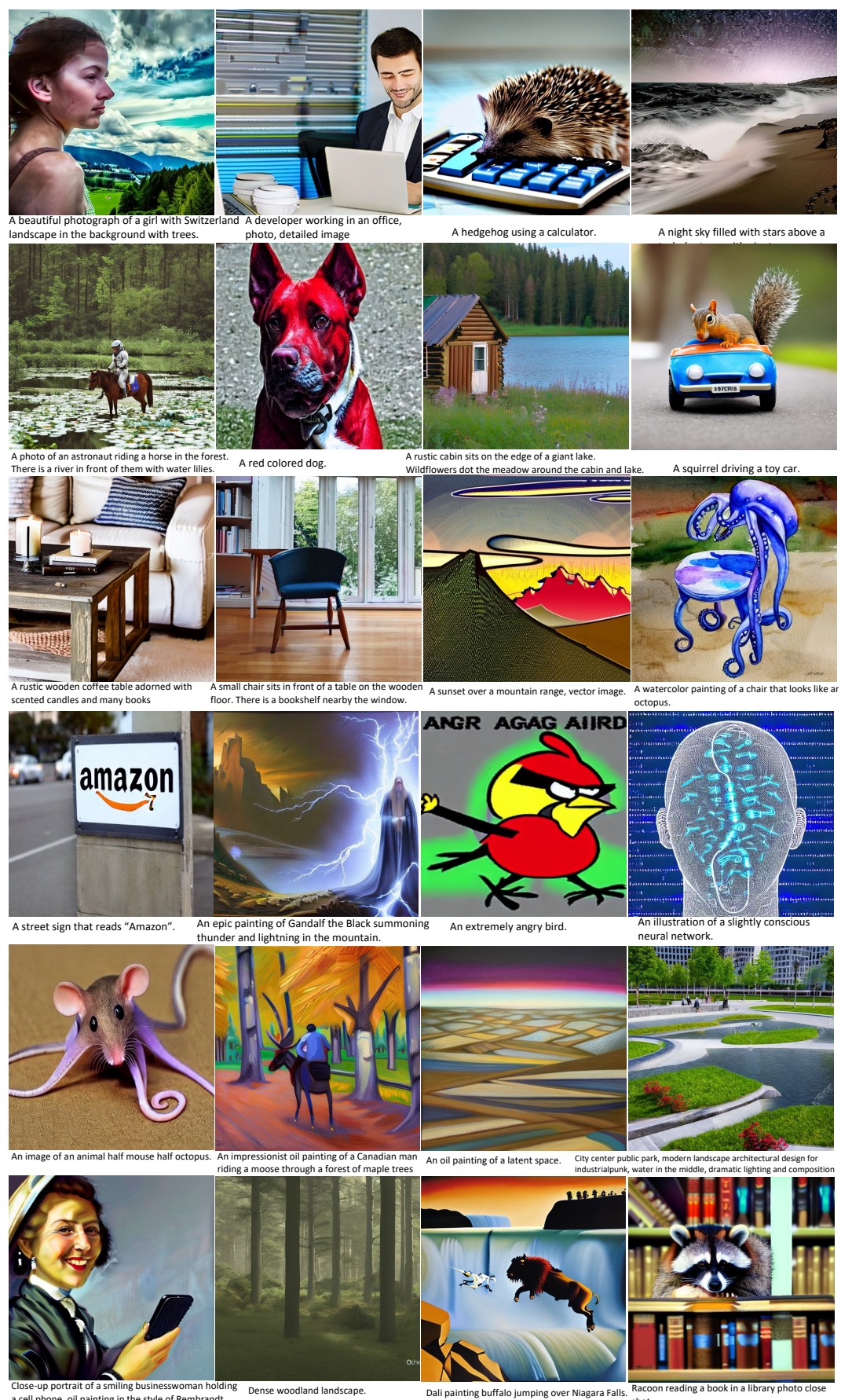

28

Figure 12: Random samples for text-to-image generation finetuned on stable diffusion v2.

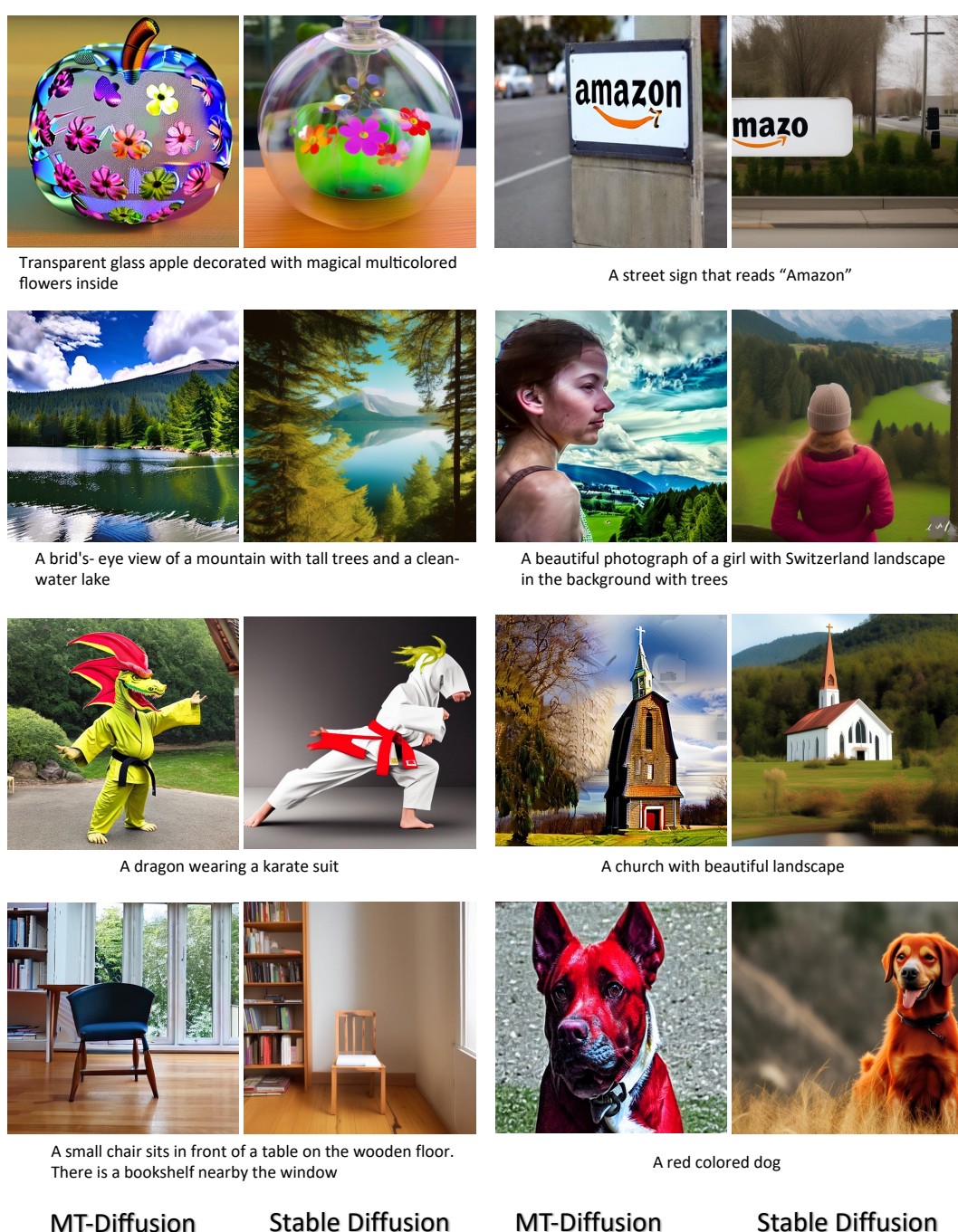

Figure 13: Visual comparisons between our MT-diffusion (left) and the stable diffusion baseline (right).

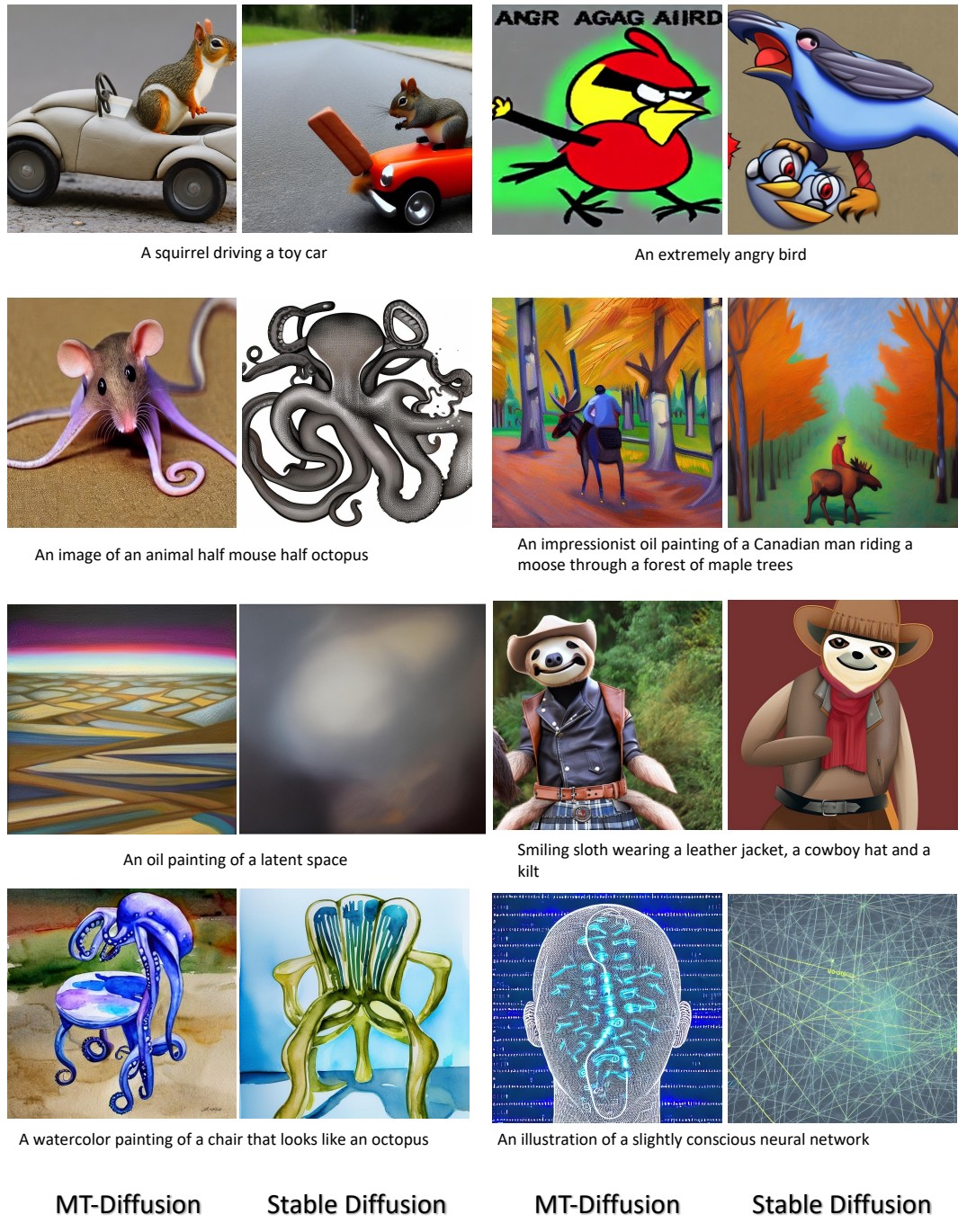

Figure 14: Visual comparisons between our MT-diffusion (left) and the stable diffusion baseline (right).

