# OpenReview forum: "Diffusion Models for Multi-Task Generative Modeling"
_ICLR.cc/2024/Conference — ICLR 2024 poster_

### Official Review · Reviewer_4Fko · 2023-10-23

**Soundness:** 3 good
**Presentation:** 1 poor
**Contribution:** 2 fair
**Rating:** 6
**Confidence:** 4

**Summary:**

In this study, four different scenarios for multi-task learning are presented to enhance the performance of image generation and classification. The authors emphasize two specific settings, namely masked-image training and joint image-label generation. Through experiments conducted on ImageNet, the paper successfully demonstrates the viability of the suggested multi-task frameworks.

**Strengths:**

- Multi-task learning in generation, the subject of investigation in this study, is a captivating and relatively unexplored area of research.

- The methods proposed in this study exhibit simplicity and effectiveness when applied to the ImageNet-64 baseline.

**Weaknesses:**

- The explanation of the implementation details (2.2.4) regarding the encoders and decoders can be perplexing, particularly in terms of how the classification label is encoded and how it is "aggregated" with other tensors.

- The theoretical portion of the paper does not provide a clear and comprehensive explanation of the proposed multi-task models, as the focus of the paper is primarily empirical. It would be beneficial to have a more detailed architectural explanation of the models designed for various multi-task settings. If space constraints are a concern, the theoretical portion can be entirely moved to the appendix.

- While the proposed method demonstrates significant improvements on the ImageNet-64 benchmark, it lacks experiments on more widely used and challenging benchmarks, as well as comparisons with newer generation models. Additionally, some auxiliary experiments in the Appendix utilize stable diffusion, which is now commonly employed as a baseline, while the primary experiments do not present any relevant results. The absence of these experiments makes it challenging to provide sufficient justification for the superiority of the proposed complex multi-task training pipeline.

**Questions:**

- Could you please elaborate on the specific encoders employed for each task and the type of data they operate on? It would be helpful to understand the label space for each task, particularly regarding how a one-hot classification vector is encoded in the latent space.

- It is unclear from my understanding what the single-task learning model for ImageNet classification, depicted in Figure 6, entails. Further clarification would be appreciated.

- To ensure the readiness of the paper for publication, it is crucial to address the concerns mentioned in the Weakness section. These issues should be at least partially resolved. Thank you for your attention to these matters.

---

> ### Author Response · Authors · 2023-11-20
> **Rebuttal**
>
> Thank you for your thoughtful review and valuable feedback. We appreciate the opportunity to address your concerns:
>
> For Weakness 1: Implementation Details (2.2.4):
> Response: We acknowledge the complexity of the implementation details in Section 2.2.4. More detailed descriptions for different task settings, including how label data is encoded, are provided in Section 4 (Experimental Part) and Appendix E. Please refer to those sections for a comprehensive understanding, which we will put effort in to make it more clear in the revision.
>
> Specifically, for label data, we utilize the original U-Net structure from the ADM model, accepting input with a noisy image and label (from an embedding layer as a conditional input). In our case, the noisy image is set as the original image, and label embeddings are jointly encoded through the U-Net.
>
> For Weakness 2: Theoretical Portion:
> Response: Thanks for the suggestions. We respectfully disagree that our work is primarily empirical. Indeed, there is no theory on how to design a diffusion model that can handel multi-modal multi-task data by incorporating different task information in the forward process like our model. We consider this as one key contribution of our work. We believe it is necessary to present the underlying theoretical perspective of our model, which will provide guidance to design specific model architectures for multi-task generative modeling. However, we agree that the presentation can be simplified and emphasizes more on the model design part. We will be happy to do a thorough revision of our paper to reflect the suggestions from the reviewer.
>
> For Weakness 3: Experiments on Widely Used Benchmarks:
> Response: We understand the importance of benchmarking on widely used datasets. In our main experiments, we train on both ImageNet-64 and ImageNet-128 based on the ADM codebase, providing a well-documented baseline with accepted evaluation scores for comparison. Training larger models on datasets like LAION from scratch is computationally challenging. More importantly, there are many factors such as the data quality in the training that can significantly impact the final model performance, thus making directly comparing with these models in a fair and faithful way difficult. We believe a quantitative comparison with the ADM baseline on ImageNet sufficiently demonstrates the effectiveness of our method.
>
> For question on what the single-task learning model for ImageNet classification, depicted in Figure 6, entails.
> Response: The single-task learning method in Figure 6 is the simple U-Net trained based on pure supervised learning, which is the most direct way to do classification.

---

> ### Comment · Reviewer_4Fko · 2023-11-22
> **Some concerns are addressed, while some are remained. I will raise the scores to borderline accept.**
>
> I acknowledge the authors' efforts in providing clarification regarding weakness 1 and weakness 2.
>
> However, I still find the writing in the current version of the paper to be unsatisfactory because of the limited space. The description of the implementation details of the method in section 2.2.4 and the experiment section remain unclear. Additionally, there are several typos and format issues in the reference section.
>
> Furthermore, for weakness 3, the absence of a comparison with the prevalent latent diffusion raises doubts about the effectiveness of the proposed method.

---

> > ### Author Response · Authors · 2023-11-22
> > **Thank you**
> >
> > Dear Reviewer,
> >
> > Thank you very much for your reconsideration of our paper and raising your score. We appreciate your time and effort.
> >
> > We know that your remaining concerns are mainly about the presentation, which we agree. It was a challenge for us to present the paper as we needed to present the motivation, underlying mathematics principles, and the abundant experiments that we have done. We should revise the paper in the final revision, to put more important things in the main text, as well as some details of the model. We will put significant efforts to this revising process.
> >
> > For comparisons with stable diffusion, we acknowledge that it is very challenging to perform a fair and convincing comparison between our model and stable diffusion on large datasets such as the LAION dataset we used. However, to resolve your concern, we have conducted an additional visual comparison experiment to compare our model and stable diffusion. The results are plotted in two new figures, Figure 13 and Figure 14 in the Appendix. It appears that our model seem to understand the underlying semantics of the images better, and generates better looking images corresponding to the prompts in many cases. Please check our revision PDF file.
> >
> > Again, we appreciate your comments and willingness to raise your score. Hope our rebuttal can further resolve your concerns.

---

### Official Review · Reviewer_oMx9 · 2023-10-30

**Soundness:** 3 good
**Presentation:** 3 good
**Contribution:** 3 good
**Rating:** 6
**Confidence:** 2

**Summary:**

This paper extends the diffusion process in a multi-task learning, with a shared diffusion space for all task data. The paper verifies its formulation in multiple variations of a two-task setting (though it should be relatively straightforward to generalise into many tasks), showing improved performance over standard single task learning baselines. The paper also lists several accompanied architecture designs for the proposed multi-task diffusion formulation, based on different choices of data domains.

**Strengths:**

Disclaimer: I am probably not the right person to review this paper. I have background in multi-task learning but have limited experience in diffusion models. The paper seems to more focus on diffusion models and have limited context in multi-task learning.

-	The problem formulation is clean and straightforward. I have no problem understanding its derivation of ELBO and loss functions.
-	The presented architectures consider multiple choices of data types.

**Weaknesses:**

I will present my concerns and weaknesses here fully based on my experience in multi-task learning.

1.	The related work and experiments are all around diffusion models with very trivial baselines and simple experiments. I understand the author shape this paper as the one of the first to explore diffusion in a multi-task learning setting. But at the same time, I saw some other papers also using the multi-task learning technique in diffusion to enforce geometric constrains, particularly in the 3d shape synthesis.  For example, Wonder3D (https://www.xxlong.site/Wonder3D/) and DreamCraft3D (https://mrtornado24.github.io/DreamCraft3D/) are two examples applying diffusion on both RGB and normal maps to improve multi-view/geometric consistency. I am aware both papers were released very recently and seem to be submitted to ICLR as well, I am just wondering how the proposed paper differentiates itself from the straightforward implementation of using multi-task learning in diffusion like bering used in these two papers, and from which both were based on the same assumption of using shared diffusion space as well?
2.	As such, I am not exactly sure how to comment and understand the performance of the proposed formulation, since the experiment setting is simple and only compare with simple single task baseline. For example, it might be more intriguing to see i) the performance / formulation without having a shared diffusion space, ii) how to compare with multi-task predictive models without any diffusion (e.g. MTAN, Adasahre, Cross-Stitch, PadNet, as multi-task learning in computer vision is an active research area.), iii) is task conflict issue in multi-task learning alleviated? What are the other benefits other than improved performance?

**Questions:**

See weaknesses.

---

> ### Author Response · Authors · 2023-11-20
> **Rebuttal**
>
> We appreciate your thoughtful feedback on our work and would like to address the concerns raised in your review. We are committed to addressing these points in our revised manuscript, providing additional clarity and context where necessary.
>
> For Weakness 1: We respectfully assert that our choice of baselines and experiments is non-trivial. ADM by OpenAI, a baseline model, stands as one of the state-of-the-art diffusion generative models. Additionally, conducting experiments on the ImageNet dataset, a large and computationally demanding dataset, is far from trivial. For instance, a single experiment on 8 A100 GPU server takes several days. We believe our experiments are well-justified given the computational demands of the tasks and the significance of the chosen baselines.
>
> Thank you for pointing out these interesting references. We want to point out that an important difference between these methods and ours is that they are essentially based on the single-task diffusion model, i.e., the underlying principle is the conditional diffusion model, with additional losses to introduce some regularizations for the 3D generation. We consider at least two differences from our work: 1) the conditional generation only models a conditional distribution, whereas ours models the joint distribution of different modality data. This allows us to perform not only joint generation, but also conditional generation as the purpose of the references, i.e., ours is a more general generation framework. 2) The addition of extra losses in the reference works is mostly ad hoc, whereas in our framework, all the losses are attributed from the new multi-task ELBO, providing a more general and principled approach for multi-task diffusion generation modeling.
>
> For Weakness 2: We acknowledge the complexity of our experimental settings, focusing on incorporating multi-task learning into diffusion models—a non-trivial task given the unique characteristics of diffusion models. We appreciate your specific questions and provide the following responses:
> 1. Performance/formulation without a shared diffusion space corresponds to the baseline of modeling tasks using independent single-task diffusion models, which we have compared with.
>
> 2. How to compare with multi-task predictive models without any diffusion: Since we focus on generative modeling, it is not exactly clear to us how to compare our model with these predictive models. However, we believe some ideas can be incorporated into our framework for further performance improvement, which we would like to leave as interesting future work.
>
> 3. Is task conflict issue in multi-task learning alleviated: As a first work, the task conflict issue alleviation was not investigated in this study, which we believe would need significant effort. THus, we consider it as an interesting avenue for future exploration.
>
> 4. What are the other benefits other than improved performance: We believe the most important benefit of our framework is that it enables multi-task generative modeling, i.e., one can model multiple generation tasks in a single model without hurting single-task performance. This can not only save training cost, but also alleviate the burden of maintaining different models for different tasks.

---

> > ### Comment · Reviewer_oMx9 · 2023-11-21
> > **Response**
> >
> > I'd like to thank for the authors' thorough explanation and clarification. I have now a better understanding of this paper as it is set apart from other straightforward multi-task learning with conditional diffusion by learning a joint distribution with a multi-task ELBO loss.
> >
> > As such, I am keen to improve my rating conditioned on the following adjustments to the paper:
> > 1. Adding this conditional distribution for multi-task diffusion formulation as an important baseline on small-scale datasets.
> > 2. Adding additional discussions / related works on multi-task learning designed for predictive models, as it's easier to be appreciated by such community.

---

> > > ### Author Response · Authors · 2023-11-22
> > > **Thank you**
> > >
> > > Dear Reviewer,
> > >
> > > Thank you very much for your time and willing to revise your score. We are confident to be able to address your concerns:
> > >
> > > 1. Adding this conditional distribution for multi-task diffusion formulation as an important baseline on small-scale datasets.
> > > Q: In response to your request, we have started a job to apply our best performed version, MT-Diffusion for masked-image training on the small CIFAR-10 dataset, as well as a conditional model to generate complete images from randomly masked images. We aim to compare our method with this conditional generation baseline. We are committed to report the results in our final revision (and will report the partially trained results by the rebuttal deadline, if the training has not finished).
> > >
> > > In additional to this, we wish to point out that our current results has already contained such a comparison on the ImageNet dataset. If you look at Table 2, "ADM (class cond)" is the label-conditioned model, whereas "MT-Diffusion-M*" and "MT-Diffusion-E*" are two of our variants that model the joint distribution of images and their labels. It is clear that our model outperforms the simple conditional diffusion baseline. We hope our results can adequately address your concerns.
> > >
> > > 2. Adding additional discussions / related works on multi-task learning designed for predictive models, as it's easier to be appreciated by such community.
> > > Q: Thank you for the suggestion. We totally agree that adding discussions/related works on multi-task learning design for predictive models is a great way to enrich our paper. We have started working on that by adding a preliminary version in the appendix of the PDF file (please check out our latest main file that includes the appendix in Section D, the red-marked part). Please note that we will keep revising this section and potentially reorganize the whole paper to move it to the main text.
> > >
> > > Thanks again for your time.

---

> ### Author Response · Authors · 2023-11-23
> **experiment update**
>
> Dear Reviewer,
>
> As the deadline for rebuttal is approaching, we just want to update with you the CIFAR-10 experiment we are running. We have run the experiment on CIFAR-10 with MT-diffusion for masked-image training for around 8 hours on a single A100 GPU, as well as the conditional diffusion model that generate images from random masked images. The experiments are still not converging. In order to catch up with the rebuttal deadline, we have used the intermediate checkpoints (at the same running time) to generate 10K images, and calculate the FID and IS scores. The results are listed as follows:
>
> |Model | IS | FID|
> |:----|:----:|:----:|
> |MT-Diffusion	|11.25		|7.33|
> |Cond-DIffusion	|10.11		|8.43|
>
> Please note since the algorithm has not converged, these results are not comparable to current state of the art. However, we believe our final results can catch up quickly. More importantly, the current results indicate our model by modeling the joint distribution performs better than the simple conditional diffusion model. We will continue the experiment and add the final results to the final revision.
>
> We appreciate your willingness to adjust your scores. Given our current and existing results, as well as our above rebuttals, can you please consider doing so? Thank you very much.

---

### Official Review · Reviewer_KnqL · 2023-11-01

**Soundness:** 2 fair
**Presentation:** 2 fair
**Contribution:** 2 fair
**Rating:** 6
**Confidence:** 5

**Summary:**

This work discusses the potential of diffusion-based models in generative modeling. While current diffusion models excel in single-generation modeling, the paper explores the possibility of extending them for multi-task generative training. The authors introduce a unified multi-task diffusion model, MT-Diffusion, that operates in a shared diffusion space. This model aggregates information from multiple types of task-data and employs a shared backbone denoising network with task-specific decoder heads. The paper presents several multi-task generation settings, such as image transition, masked-image training, joint image-label, and joint image-representation generative modeling. Experimental results on ImageNet demonstrate the model's effectiveness in multi-task generative modeling.

**Strengths:**

- The paper introduces MT-Diffusion, a novel approach to multi-task generative modeling using diffusion models.
- The proposed model effectively aggregates information from different task-data types, enhancing its versatility.
- Extensive experiments on ImageNet validate the model's effectiveness and potential in various multi-task generative modeling scenarios.

**Weaknesses:**

- Experiments are done on low resolution and small datasets, undermining its effectiveness.
- The paper is lack of model details for each task

**Questions:**

- Can you explain the model structure for each task and result? Especially results in Fig 4 and Fig 5.

---

> ### Author Response · Authors · 2023-11-20
> **Rebuttal**
>
> Thank you for your thoughtful evaluation and constructive feedback. We appreciate the opportunity to address your concerns:
>
> For Weakness 1: Experiments on Low Resolution and Small Datasets:
> Response: We respectfully disagree with the characterization of our experiments. While the primary experiments were conducted on the ImageNet1K dataset, which is widely acknowledged as substantial in the generative modeling community, we acknowledge the existence of larger datasets such as LAION. However, due to the prohibitive cost of pretraining from scratch on these datasets, they are not considered standard for comparisons.
>
> We conducted experiments on two resolution levels, 64x64 and 128x128, leveraging our computational resources. These experiments, although time-consuming, are deemed sufficient to demonstrate the effectiveness of our method. Additionally, qualitative experiments on the larger LAION dataset, based on the pretrained stable diffusion model, are detailed in the appendix for further insights.
>
> For Weakness 2: Lack of Model Details for Each Task:
> Response: We appreciate your observation, and we want to clarify that all model details are provided in the experiment section and Section E of the Appendix, adhering to page limits. To enhance clarity, we are willing to paraphrase the paper and incorporate more detailed experimental settings into the main text.
>
> For the Question on Model Structure:
> Response: Detailed information on experimental settings and model structures is provided in Appendix E. The model structures closely resemble the ADM model, incorporating a shared U-Net backbone shared by all tasks, as illustrated in Figure 3. Task-specific encoder structures are introduced in our model for added specificity. For a comprehensive understanding, please refer to Appendix E, and feel free to reach out with any additional questions.

---

> > ### Comment · Reviewer_KnqL · 2023-11-21
> > **Thanks for response**
> >
> > Thank you for your response.
> >
> > After reviewing the rebuttal and the supplementary materials, I decide to increase my score.

---

> > > ### Author Response · Authors · 2023-11-21
> > > **Thank you**
> > >
> > > Dear Reviewer,
> > >
> > > We appreciate your recognition of our work by raising your score. Thank you very much for your comments. We will definitely revise our paper by taking all you comments into consideration.

---

### Official Review · Reviewer_bBAj · 2023-11-08

**Soundness:** 3 good
**Presentation:** 3 good
**Contribution:** 2 fair
**Rating:** 5
**Confidence:** 4

**Summary:**

This paper introduces a novel approach to generative modeling by extending diffusion-based models to a multi-task learning framework. The proposed Multi-Task Diffusion Model (MT-Diffusion) is capable of generating multi-type data (e.g., images and their corresponding labels) within a single unified model. It integrates multi-task learning losses into the diffusion process, supported by a theoretical foundation. The authors propose and experiment with several multi-task generative settings, including image transition, masked-image training, joint image-label, and joint image-representation generation, demonstrating the framework's versatility and effectiveness on the ImageNet dataset. MT-Diffusion handles multiple data types through a shared diffusion space, with a forward process aggregating multi-task data and a reverse process using task-specific decoder heads to reconstruct data for different tasks. This approach results in a novel multi-task variational lower bound that generalizes the standard diffusion model, achieving simultaneous multi-task generation without compromising individual task performance.

**Strengths:**

- The paper provides a sound theoretical explanation for the utility of a multi-task loss using the Evidence Lower Bound (ELBO).
- The idea of enabling multi-task learning for inputs of various modalities through a shared latent space is innovative.
- Considering the connection to guided diffusion models is a thoughtful approach that takes into account the expansiveness of the research.

**Weaknesses:**

- The paper does not specify the extent of increased training costs resulting from the proposed methodology.
- While significant performance improvements are shown across various metrics, including FID, the analysis lacks control of variables to confirm that these improvements truly stem from a multi-task setting. Following the previous point, it is my view that the proposed methodology likely entails considerably higher training costs and an increased number of data samples seen by the model compared to baseline learning. Therefore, it is necessary to deeply analyze whether the performance improvement is due to positive transfer resulting from multi-task learning, or merely an effect akin to data augmentation from masked samples. The absence of such analysis has influenced my evaluation towards rejection.
- (Minor point) There is prior (possibly concurrent) work proposing a multimodal, multi-task diffusion process through a Versatile Diffusion[1] multi-flow diffusion process.
- (Minor point) The caption of Table 2 does not provide sufficient information, making it difficult for the reader to interpret the experimental results.

[1]: Xu, Xingqian, et al. "Versatile diffusion: Text, images and variations all in one diffusion model." Proceedings of the IEEE/CVF International Conference on Computer Vision. 2023.

**Questions:**

- How do you think the proposed off-the-shelf guidance method would integrate with previous research focused on efficient training of diffusion models, such as P2-Weighting[2], Min-SNR[3], ANT[4], and Task Routing[5]? Particularly, Min-SNR[3], ANT[4], and Task Routing[5] view the methodology of diffusion models as naturally creating a multi-task situation through various time steps, each requiring different levels of denoising. Considering this study aims to extend multi-task learning by increasing the input modality for denoising, there seems to be an overlap. I would be interested to hear your insights on this matter.


[2]: Choi, Jooyoung, et al. "Perception prioritized training of diffusion models." Proceedings of the IEEE/CVF Conference on Computer Vision and Pattern Recognition. 2022.

[3]: Hang, Tiankai, et al. "Efficient diffusion training via min-snr weighting strategy." arXiv preprint arXiv:2303.09556 (2023).

[4]: Go, Hyojun, et al. "Addressing Negative Transfer in Diffusion Models." arXiv preprint arXiv:2306.00354 (2023).

[5]: Park, Byeongjun, et al. "Denoising Task Routing for Diffusion Models." arXiv preprint arXiv:2310.07138 (2023).

---

> ### Author Response · Authors · 2023-11-20
> **Rebuttal**
>
> Thank you for your thoughtful review and valuable feedback. We appreciate the careful consideration given to our work. Below, we address each of the raised points and provide additional clarification.
>
> For Weakness 1: Increased Training Costs:
> We appreciate your concern regarding potential increased training costs. In our two-task setting, the additional computational overhead is minimal, amounting to approximately 10% more time per iteration than a pure single-task diffusion model. Importantly, our model remains significantly more efficient than training two individual diffusion models for the two tasks separately in terms of both time and storage efficiency. We'd like to highlight that our method not only handles multi-task learning effectively but also converges faster, as indicated in Table 2. We will incorporate this discussion into the final revision for enhanced clarity.
>
> For Weakness 2: Where performance improvement comes multi-task learning:
> We wish to argue that the additional data augmentation in the masked image training is also a part of the multi-task setting, offering additional benefits without incurring extra overhead. We believe separating data augmentation from the multi-task setting is challenging, as that is part of the multi-task framework.
>
> We do not fully understand your point about positive transfer and data augmentation, because the auxiliary task only contains data augmentation. Thus, we believe the data augmentation indeed attributes to the performance gain, which can also be understood as positive transfer. We will explicitly discuss the insight in the revised manuscript.
>
> For Weakness 3: Comparison with Prior Work:
> Thank you for bringing up the reference to Versatile Diffusion. After a careful review, we recognize that it is indeed a concurrent work that we did not initially notice. The Versatile diffusion mainly focuses on developing new neural architectures that make different tasks interact with each other within the single-task diffusion framework. We believe that our work is different, as it not only introduces a novel neural architecture but also generalizes the single-task diffusion in the loss function. We will enrich the discussion on the relationship between our model and Versatile Diffusion in the revision.
>
> For Weakness 4: Table 2 Caption:
> We appreciate your feedback on the clarity of the Table 2 caption. We acknowledge that additional details about different variants of our model are described in Sections 4.1 and 4.2. To enhance clarity, we will provide more comprehensive information in the title for improved interpretability in the revised manuscript.
>
> For Questions: Relation with Previous Research:
> We are grateful for the references to inspiring papers interpreting standard diffusion models from a multi-task learning perspective. While we acknowledge the relevance of these papers, we maintain that our ideas are orthogonal, with no technical overlap. We are open to exploring the incorporation of these insights into our framework for potential further improvement—a promising avenue for future work.
> Thank you once again for your thoughtful review. We are committed to addressing each of these points thoroughly in the final revision, ensuring a more robust and comprehensive presentation of our work. We hope the reviewer can reconsider his/her decision.

---

> ### Comment · Reviewer_bBAj · 2023-11-21
>
> The concerns I have raised have been largely addressed, and I appreciate the efforts made by the authors in this regard. However, my concern regarding Weakness 2 remains unaddressed. I acknowledge that it may be challenging to completely disentangle the effects of data augmentation from multi-task learning. Nonetheless, it is imperative to support, through experimental or theoretical analysis, whether the proposed multi-task scenarios actually synergized to enhance performance. For instance, assessing task affinity to verify positive transfer between tasks would significantly bolster the validation of this research's utility.

---

> ### Author Response · Authors · 2023-11-21
> **Thank you for your feedback**
>
> Dear Reviewer,
>
> We appreciate your feedback on our rebuttal. We are happy to learn that you are mostly satisfied with our rebuttal, and only have some concerns on verifying whether the performance comes from positive transfer. We are happy to explore and performance more ablation studies on this. Specifically, we have set up to run the following experiments: train our multi-task diffusion model on 5 tasks to simultaneously learning to generate original images,, masked images, corresponding captions, random captions, and class labels. Intuitively, generating original images and masked images are two closest tasks, which is expected to have more positive transfers. While the other tasks such as generating random captions are completely independent of generating images, thus we expect the performance would drop if the task of generating random captions are added. We are working on the experiments now, but please note this is a more challenging setting, thus we are not expecting to be able to produce some results before the rebuttal deadline. However, we are committed report the results in our revision.
>
> Even though we are not able to provide the aforementioned results in the rebuttal period ending in one day, we wish to point out our current results actually have the implications that more similar tasks tend to have more positive transfers. For example, comparing the following two settings indicated in Table 2: 1) simultaneously generating images and the corresponding masked images; 2) simultaneously generating images and the corresponding labels. The former two tasks are considered closer as they are in the same data space. And from Table 2, we can clearly see that the former setting (Generation with Masked-Image Training (Section 4.1)) outperforms the latter one (Generation with Joint Image-Label Modeling (Section 4.2)), indicating there are more positive transfers in the former setting.
>
> We hope this can address your concerns, and our explanation can provide more information on your final decision. Thank you very much for your time.

---

> > ### Author Response · Authors · 2023-11-23
> > **Follow up**
> >
> > Dear Reviewer,
> >
> > Thanks again for your time reviewing our paper. We just want to send out a follow-up request to please reconsider your final decision of our paper. Specifically, we believe our existing results, as pointed our above, can at least partially resolve your concern on the positive transfer and task similarity issue. Furthermore, we have setup an extra experiment to verify that. Due to the complexity of the experiment, we cannot provide results during the rebuttal period. However, we are committed to finish that in our final revision.
> >
> > Furthermore, we would like to highlight that we have provided some additional experiment verifications to address the issues raised by other reviewers, which are reflected in the Appendix of the updated PDF file. As a result, all other reviewers agree to raise their scores.
> >
> > Again, thanks a lot for your time and effort in reviewing our paper. We appreciate it if you can adjust your final decision.

---

### Author Response · Authors · 2023-11-20
**Rebuttal**

Dear Reviewers,

Thank you very much for the thoughtful reviews. After carefully reading the reviews, we believe the reviewers' concerns are mostly about personal preference instead of technical flaws. We have addressed the concerns in the individual rebuttals. We appreciate your time and hope the reviewers can reconsider their evaluations based on our rebuttal.

---

### Meta-Review · Area_Chair_ZN3J · 2023-12-06

**Metareview:**

- Claims and findings:
This submission proposes a multi-task learning framework for diffusion models, with a shared space for different tasks. The formulation is tested in different two-task settings where experimental results improved performance over single-task settings. In addition, the submission also adds several architectural designs based on different data domains.


- Strengths:

The paper poses an interesting question, mainly whether training a diffusion model in a multi-task setting is possible and what's the expected performance of such approach. Reviewers have highlighted that the paper provides a sound theoretical explanation for the utility of a multi-task loss using the Evidence Lower Bound (ELBO). In addition, the idea of enabling multi-task learning for inputs of various modalities through a shared latent space is innovative.

- Weaknesses:

Reviewers have noted that the experimental setup is composed of trivial baselines and simple experiments. In addition, it's unclear how some of the architectural settings differ from those used in 3D generation.

- Missing from submission:

Reviewers have pointed out that a complete evaluation pipeline with more complex tasks and recent baselines could be beneficial to tease out the contribution. I believe this is true to some extent but I think the paper still deserves some exposure to the ICLR community.

**Justification For Why Not Higher Score:**

There are some concerns about the strength of the experimental settings. Stronger baselines and more substantial multi-task problems could have been used.

**Justification For Why Not Lower Score:**

The multi-task diffusion model setting is novel and I believe it deserves visibility in the community.

---

### Decision · Program_Chairs · 2024-01-16

Accept (poster)